# Feeling moved by music: Investigating continuous ratings and acoustic correlates

**Jonna K. Vuoskoski** [1,2,3] *, **Janis H. Zickfeld** [4], **Vinoo Alluri** [5], **Vishnu Moorthigari** [5], **Beate Seibt** [2]

**1** RITMO Centre for Interdisciplinary Studies in Rhythm, Time and Motion, University of Oslo, Oslo, Norway, **2** Department of Psychology, University of Oslo, Oslo, Norway, **3** Department of Musicology, University of Oslo, Oslo, Norway, **4** Department of Management, Aarhus University, Aarhus, Denmark, **5** Cognitive Science Lab, International Institute of Information Technology, Hyderabad, India

☯ These authors contributed equally to this work.
* j.k.vuoskoski@imv.uio.no

**Data Availability Statement:** All data, analysis scripts and materials are available at our project page: https://osf.io/xgj85/.

## Abstract

The experience often described as feeling moved, understood chiefly as a social-relational emotion with social bonding functions, has gained significant research interest in recent years. Although listening to music often evokes what people describe as feeling moved, very little is known about the appraisals or musical features contributing to the experience. In the present study, we investigated experiences of feeling moved in response to music using a continuous rating paradigm. A total of 415 US participants completed an online experiment where they listened to seven moving musical excerpts and rated their experience while listening. Each excerpt was randomly coupled with one of seven rating scales (perceived sadness, perceived joy, feeling moved or touched, sense of connection, perceived beauty, warmth [in the chest], or chills) for each participant. The results revealed that musically evoked experiences of feeling moved are associated with a similar pattern of appraisals, physiological sensations, and trait correlations as feeling moved by videos depicting social scenarios (found in previous studies). Feeling moved or touched by both sadly and joyfully moving music was associated with experiencing a sense of connection and perceiving joy in the music, while perceived sadness was associated with feeling moved or touched only in the case of sadly moving music. Acoustic features related to arousal contributed to feeling moved only in the case of joyfully moving music. Finally, trait empathic concern was positively associated with feeling moved or touched by music. These findings support the role of social cognitive and empathic processes in music listening, and highlight the social-relational aspects of feeling moved or touched by music.

## Introduction

The emotional effects of music are among the most important reasons for engaging in music listening in everyday life (e.g., [1–3]). These effects range from slight changes in affective state to exceptionally strong, transformative experiences [4]. One commonly reported response to

**Funding:** JKV: This work was partially supported by the Research Council of Norway through its Centres of Excellence scheme, project number 262762 https://www.forskningsradet.no/en/ The funders had no role in study design, data collection and analysis, decision to publish, or preparation of the manuscript.

**Competing interests:** The authors have declared that no competing interests exist.

music is feeling moved or touched [1]. Beyond the domain of music, this phenomenon has garnered increased interest in recent years. Theoretical arguments and empirical evidence suggest that people often say they are moved or touched in response to increased affiliation and morality, and that the emotional state is experienced as predominantly positive, often features tears, chills, or warm feelings, and motivates social bonding [5, 6] (Zickfeld et al., 2019). This evidence further indicates that people across a vast array of cultures and languages respond similarly to elicitors of the emotional state, labeling their state with corresponding terms in their language. In English, people typically use the terms *moved* and *touched* to describe their state. We will use *feeling moved* to denote this subjective feeling state.

However, there exist few systematic studies that have tested the convergence of these features in response to music. Rather, prior studies focused on specific components or certain types of music only (e.g., [7]). Are people moved by the same affiliative aspects when listening to music as they are when for example reuniting with a loved one? What musical properties facilitate feeling moved? There is some evidence that music can indeed convey affiliation motives and that listeners can feel socially connected to different aspects of music or instruments [8–10].

In the present study, we explore correlates of feeling moved or touched in response to music, focusing on the question of whether theories of feeling moved can account for responses to music and which musical and acoustical features contribute to the experience of feeling moved. Instead of probing individuals' subjective responses after listening to different musical excerpts, we asked participants in the current study to continuously rate their experiences and perceptions while they unfold during music listening using a continuous self-report paradigm (e.g., [11]).

## Theories of feeling moved

Building on anthropological and ethnographic work, Fiske et al. [12] introduced *kama muta* as a universal tendency to respond emotionally (in a way that is often described as moved or touched) when communal sharing suddenly intensifies. Communal sharing is one of four basic schemes (or models) of relating to others focusing on what we have in common. It is expressed by giving according to need and ability (while the other three models are expressed by giving according to hierarchy, equality and proportionality, respectively), by bodily proximity, touch, synchrony, or food sharing (see [13], for a more detailed introduction of communal sharing and the remaining models). States of kama muta are thus characterized by appraisals of increased interpersonal closeness or connectedness (reflecting the intensification of communal sharing), by labeling them as moved, touched or heartwarming, as well as by sensations of tears, warmth in the chest and chills, by experiencing them as positive, and by renewed devotion to one's relationships characterized by communal sharing [14, 15].

Empirical evidence has supported kama muta theory across several different cultural contexts [5, 16]. However, so far the empirical studies testing kama muta theory have only used stories and videos to evoke kama muta through the intensification of communal sharing between human characters. Fiske [14] assumes that the same appraisal of increased closeness or connectedness also accounts for instances where people describe being moved by music: "Sometimes the CS that suddenly intensifies is between musicians and audience, sometimes among the musicians or singers, sometimes among the audience, sometimes with the composer or even with the music itself." However, so far, this proposition has not been tested.

Kama muta theory defines the emotion of kama muta through the main appraisal theme, intensification of communal sharing (labelled an *elicitor-specific* eudaimonic emotion in recent theorizing, [17]). The labels that persons give their emotional experience, such as moved,

touched, heartwarming, raptured and others, in English, are but one, though for diagnostic purposes often very important, index for the emotion [12].

Conversely, other theories define the emotion through the label "being moved", encompassing assumed elicitors, subjective feelings, physiological signs and sensations and often also motivational tendencies (a *feeling-specific* eudaimonic emotion according to Landmann [17], see [6], for a review). Of these, two are particularly relevant for the present investigation. Konečni [18, 19] treats being moved as part of the *aesthetic trinity*, next to *thrills* or *chills* and states of *aesthetic awe*. Thereby, being moved represents strong emotional states that are experienced in response to the *sublime* (including musical stimuli). He contends that chills are, in the context of music, far more frequent and predictable (because they are shallower) than the state of being moved. He considers being moved to be a rarer response and determined by the personal associative context of the person. If this was the case, there should be low interindividual agreement on what the moving segments in a musical piece are. Conversely, kama muta theory suggests that being moved can be evoked reliably by musical passages because the music itself conveys an intensification of communal sharing. We also expect that for moving music, the occurrence of chills and feeling moved or touched coincide to a large degree.

The other theory we draw on for the present investigation is the *distancing-embracing* model [20], which posits that being moved is involved in transforming negative states into pleasurable experiences in aesthetic perceptions, including music listening. Being moved is thereby conceptualized as a mixed emotion that is experienced as predominantly positive. The authors distinguish between two prototypes of being moved: a sadly moving and a joyfully moving variant [21, 22]. They expect that being moved plays a more important role in the enjoyment of *sad* stimuli. Indeed, empirical research has indicated that being moved is a mediating factor in explaining the enjoyment of artworks that are experienced as *sad* [22, 23]. Similar findings have also been obtained for sad music, and empathic responses were identified as a possible driver [7, 24].

Empathy, in turn, has been identified as a characteristic of at least an important variant of being moved, and empathic predispositions predict the propensity to feel moved [6, 21, 25]. It remains to be shown, however, whether feeling moved or touched by predominantly sad versus joyful pieces of music have the same profile, which would strengthen the argument that it is the same emotion in different affective contexts.

To summarize, based on kama muta theory we predict that feeling moved or touched by music co-occurs with a sense of connectedness and weeping, feelings of warmth in the chest, and chills. While these attributes are compatible with all theories on the state of feeling moved (for overviews, see [6, 26]), the aesthetic trinity theory [18] predicts low agreement regarding which segments move individuals, while kama muta theory predicts high agreement for music that is preselected to be experienced as moving. This high agreement should be caused by similar emotional responses to musical features, and we shall attempt to identify such features. Lastly, we predict that feeling moved by sadly and joyfully moving music will show comparable correlation patterns. This would suggest that it is the same emotion in both cases, whether two variants of the same mixed emotion [21] or the same emotion with different concurrent emotions of sadness and joy [14].

## Feeling moved as a response to music

As briefly alluded to earlier, feeling moved has received increased theoretical and empirical attention in the context of music. In a questionnaire study a total of 141 participants indicated that feeling moved was their fourth most common emotional response in listening to music [1], and interviews reveal that individuals often say that they are moved in the context of

strong and profound experiences with music [4]. Similarly, Scherer and Zentner [27] identified feeling moved as a common response when listening to music, involving symptoms such as moist eyes or chills (although they treated feeling moved as a *vague emotional category*) and a recent survey showed that feeling moved is the most commonly reported emotional state when investigating crying in response to music [28]. Items assessing feeling moved have been included in the Geneva Emotional Music Scale [29] and a hierarchical clustering approach suggests that feeling moved (as measured by the items moved and touched) form a distinct cluster of musically induced emotions [30], being most similar to the states of wonder and transcendence.

In a recent review of factors affecting the enjoyment of sad music, Eerola et al. [24] identified feeling moved as a possible mediator, next to empathy or social surrogacy. Across two experiments Vuoskoski and Eerola [7] examined this proposition empirically. Testing 327 participants across two experiments, the authors presented several musical excerpts that differed in terms of *movingness* and *sadness*, and collected responses on felt sadness, feeling moved and liking. In the first experiment, feeling moved fully mediated the relationship between felt sadness and liking (see [23] for similar findings using film stimuli), and the second experiment further confirmed that this relationship could not be explained by the perceived beauty of the musical pieces. Rather, movingness appeared to contribute to the perceived beauty of sad music. Given these theoretical and empirical findings, Zickfeld [31] suggested that kama muta theory might incorporate and explain several factors contributing to the enjoyment of sad music (identified in Eerola et al.'s [24] review): feeling moved, empathy, social surrogacy, or more precisely communal sharing and affiliation, and experiencing sad music as positive and pleasurable.

While it is easy to argue that music may increase affiliation through lyrics expressing tenderness and love [12], the ways in which instrumental music is to convey affiliation seem less straightforward. On one hand, a listener might experience an increase in affiliation by identifying or feeling a connection to the music in general (for example due to rhythmic entrainment; see e.g., [32]), the composer, the musicians, or other fans/listeners. Listening to familiar music may also remind us of nostalgic relationships with significant others: Konečni [18] argues that feeling moved by music is individually determined by the associative context or *web* that one has constructed with the musical piece, possibly also involving an affiliation towards certain aspects of the music. Evidence from an empirical study on nostalgia responses to music supports this assertion [33]. The authors found that musical pieces were rated as more nostalgic if they were perceived as familiar or more autobiographically salient. In addition, there is evidence that listeners can infer affiliative motives from musical improvisations [8], can feel connected to music through a form of empathy [34], and that listening to moving music from a specific culture can increase affiliation towards that culture [35].

On the other hand, in more interactive contexts feeling moved might occur due to concert attendants dancing or moving in synchrony, singing in unison, as well as perceiving the musicians performing music in synchrony. All of these aspects have been shown to elicit social bonding (e.g., [36]), and are part of communal sharing relationships that might intensify in specific contexts [13].

**Musical chills.** Further, strong emotional responses to music have been associated with the occurrence of chills, goosebumps or *frissons* [37–42]. Chills are considered a relatively common, positive psychophysiological response to music or aesthetic stimuli (see [37], for a review). Research has sometimes distinguished between chills as a subjective emotional response and goosebumps or piloerection as an objective physiological response [43]. In the present manuscript, we follow Maruskin et al. [44], who argued that goosebumps represent one physiological component of pleasurable chills.

Empirical studies have linked the occurrence of chills to strong experiences of feeling moved using qualitative methods, self-report ratings and more observational techniques such as camera devices [5, 22, 38, 43, 45]. Bannister [39] found in a comprehensive survey on chills responses to music that a high number of participants highlighted social aspects such as feelings of connectedness, the human voice, and perceived relationships between virtual agents (i.e., musical instruments/parts) as evoking chills. Similarly, a recent survey study identified *moving chills* as one of three types of chill responses to aesthetic stimuli, which co-occurred with feeling moved, tenderness, and tears (the other types being *warm chills* and *cold chills*, Bannister [38]; see also [44]). Such results fit previous findings and theories on feeling moved and highlight the importance of chills as a correlate of feeling moved or touched in response to music.

## Feeling moved and musical features

To the best of our knowledge, there has been no direct research linking responses of feeling moved by music to specific musical features (though this has been done for films; [43]). However, indirect evidence focusing on physiological responses to music such as chills or tears, symptoms that are also strongly associated with feeling moved, has been provided. Sloboda [46] found that tears were most often associated with appoggiaturas, while shivers or chills were linked to excerpts containing new or surprising harmonies (see also [47]). Later and Panksepp [42] identified solo instruments that emerge from an orchestral background and crescendos as elicitors of chills. Such dynamic changes in loudness have also been found to be an indicator of chills in more recent research [40, 48, 49]. Relatedly, an experimental study found that increased loudness (acoustic intensity) and reduced brightness (proportion of high to low frequencies) resulted in more frequent reports of chills [39]. Further research has also identified the entrance of a voice and surprising changes or violations of expectations as structural elicitors of chills [47]. As Grewe et al. [40, 47] noted, there seems to be no consistent evidence for a specific chill-inducing acoustical pattern. Rather, interactions among different musical features might be more successful in eliciting a chill response, possibly also depending on additional psychological variables at the individual level such as empathy or familiarity.

## The present study

Not only music performance but also music perception seems to be inherently social. Listeners can infer affiliative motives from musical improvisations [8] and feel connected to music through a form of empathy [34], and music can increase connectedness towards aspects conveyed and embodied by music [35]. While there exists ample evidence that feeling moved often constitutes strong emotional responses to musical stimuli [1, 4], the underlying mechanisms or conditions that moderate this reaction are yet to be investigated. As states of feeling moved typically occur in response to significant social or communal events, we argue that the inherent affiliative signals in music can evoke this particular emotion. Listeners can feel connected to aspects of the music, aspects of the performer(s), aspects of co-listeners, or an interaction of these variables [31].

In the present study we aim to investigate experiences labeled as feeling moved in response to music using continuous self-reports, a common paradigm in research on music and emotions that allows to model listeners' responses to music dynamically (see [11, 50, 51]). Instead of prompting participants to rate their emotional reactions *after being exposed* to a stimulus, continuous self-report paradigms collect responses while participants are presented with a stimulus. This methodology has two obvious strengths compared to self-report ratings after stimulus presentation, and extends and goes beyond previous studies on the role of feeling

moved in response to music [7, 52]. First, it allows us to obtain a more nuanced picture of the psychological dynamics and mechanisms unfolding while listening to music. It enables us to test whether the emotion components of feeling moved that have been found to co-occur in other contexts [6] also apply to music listening. For example, previous research has found a positive correlation between feeling sad and feeling moved by music [7]. However, this relationship lacks time specificity: It is not known whether some parts of the music are perceived as sad and others as moving, or whether the same parts are perceived as sad and moving at the same time. Second, it also allows us to test how experiences of feeling moved unfold based on specific musical features. As specified earlier, there is no direct evidence with regard to what musical or psychoacoustic features are associated with feeling moved.

In the present project we employed a continuous self-report paradigm based on the set-up of a recent empirical study exploring continuous ratings of different aspects of feeling moved in response to videos [53]. Participants were presented with a Likert-type scale and asked to change their ratings when their experiences and perceptions changed while listening to the musical excerpts. Seven variables or ratings were assessed continuously for all excerpts (although each participant rated only one variable per excerpt) to target the feeling, physiological, and appraisal components of the state of the emotion. To assess the feeling component, we included an item of *feeling moved or touched* as used in a previous study [53]. For the physiological component, we included items on *chills* and feelings of *warmth (in the chest)* (based on Schubert et al. [53]). The quality of warmth in the center of the chest has been strongly associated with feeling moved (e.g., [5, 53]), while there has been only limited research exploring the reaction in response to musical stimuli (e.g., [38]). At the moment it is not clear whether these feelings of warmth in the chest are associated with an actual objective temperature increase, or whether participants embody the metaphor of communality and feel social warmth (see [6, 54, 55]).

Note that we did not assess the common response of tears or moist eyes continuously (this was rated only after the music excerpt had ended), since our pilot study indicated a rather low occurrence of self-reported tears or moist eyes in response to our stimuli. For the appraisal component, we wanted to assess the affiliative signals or communal sharing aspects conveyed by the music. While previous studies have measured this aspect using either items with regard to how close someone feels to another target [5], how close the protagonists are to each other, or with the inclusion-of-the-other-in-the-self scale measuring closeness by showing two circles increasing in overlap [53, 56], we felt that it might seem artificial to probe how close someone feels to a certain musical piece while it unfolds. Based on the findings of a pilot study, we decided to assess the extent to which the participants felt a sense of connection when listening to music, relating to findings that individuals are able to detect affiliative intentions in instrumental musical stimuli [8].

Given that moving music is perceived as more beautiful than non-moving music [7, 28], we also included an item on perceived beauty in order to test whether continuous ratings of beauty are similar to ratings of feeling moved. Finally, in line with previous studies, we also assessed how *sad* and *joyful* participants perceived the music. Specifically, we targeted the emotions perceived in the music and not felt emotions for these two items, although it should be noted that the two processes are strongly related (see [7]). This choice was made due to the possibility that the continuous ratings of experienced joy could be confounded by enjoyment of the music as a result of being moved by the music. We were, however, more interested in the elicitors of feeling moved or touched.

Some theorizations have distinguished between the categories of being *sadly* and *joyfully moved* [21], and have found different predictions for a number of variables (e.g., [22]). Although all theories on feeling moved emphasize that the emotion is experienced as primarily

positive, they do not exclude the possibility that it occurs together at the same time with negative emotions such as sadness. Thus, in the present study we focused on including musical excerpts that were rated as both *moving* and *sad*, as well as both *moving* and *joyful* in order to explore possible diverging patterns.

Based on previous findings and theories, specifically kama muta theory, we derived a number of different predictions that were pre-registered prior to conducting the study (https://osf.io/76adr). We predicted that the time course of:

1. feeling moved or touched (feeling component) correlates positively with the time course of experiences of warmth in the chest, experienced chills (physiological component), and experiencing a sense of connection (appraisal component) across all excerpts;

2. feeling moved correlates positively with the time course of perceived beauty across all musical pieces;

3. feeling moved correlates positively with the time course of perceiving the music as sad for *sadly moving* excerpts, while it does less so for joyfully *moving* excerpts;

4. feeling moved correlates positively with the time course of perceiving the music as joyful for joyfully *moving* excerpts, while it does less so for *sadly moving* excerpts.

In addition, we explored relations between feeling moved and musical or acoustic features of the music. As there exists only indirect evidence based on musical chills, we did not pre-register any specific hypotheses but rather treated this aspect of the study as exploratory. Finally, we explored the relationship between ratings of feeling moved and two facets of trait empathy [57]; empathic concern and fantasy. Fantasy denotes the tendency to imaginatively transpose oneself into the feelings and experiences of fictitious characters, and it has been found to predict feeling moved by sad music in prior studies [7, 52]. Empathic concern denotes the sympathy and compassion one feels for others in need. It has been found to predict feeling moved across several studies and cultures and 3000+ participants with an overall effect size of $r = .35$ [5, 58].

## Materials and methods

The study was approved by the internal ethics committee of the Department of Psychology at the University of Oslo. In addition, we pre-registered the main parts of the study (https://osf.io/76adr/). Deviations from the pre-registration protocol are denoted explicitly as such. All data, analysis scripts and materials are available at our project page (https://osf.io/xgj85/).

### Participants

Previous studies employing continuous self-report paradigms have not considered sample size justifications systematically [11, 50, 51]. As an exception, a recent methodological paper considered between 20 and 30 raters per cell (combination of independent variables) to be sufficient, though this number was dependent on the specific study design [59]. We originally registered to collect 40–50 participants per cell based on a previous study using a similar procedure [53]. Performing a post-hoc precision analysis on our data following McKeown and Sneddon [59] suggested that this number was sufficient (S5 Fig in S1 File). As we had 49 cells in total (7 excerpts x 7 ratings) and each participant completed seven of these, we aimed for at least 350 participants. Due to exclusions five of the 49 cells had fewer than 40 participants in the final dataset (see S4 Table in S1 File).

In total, 423 participants were recruited on Amazon MTurk, requesting only workers with at least 95% approval rating for tasks on Amazon MTurk and location set to the US. All

participants were asked to provide informed consent before participating. After applying the pre-registered exclusion criteria, the final sample consisted of 415 US American participants (197 women, 183 men, 35 unspecified gender) ranging from 19 to 68 years of age ($M_{age}$ = 36.05, $SD_{age}$ = 10.52). Responses to an excerpt were excluded if participants spent less than two minutes on the page presenting the excerpt, which was recorded with a timer, thus resulting in some participants completing less than seven excerpts and ratings. In addition, participants were excluded if they indicated an age younger than 18, a different nationality than the US, if more than 50% of the questions were unanswered, or if they failed a probe item that assessed whether they understood the instructions. We excluded non-US participants to ensure that the participants had a shared, consistent understanding of the emotional labels and concepts used in the rating tasks, and to control for possible effects of cultural background on emotional experiences in response to music (cf. [60]).

## Procedure

The present experiment followed a 7 (within: musical excerpt) x 7 (within: rating scale) mixed design. Each of the seven excerpts was randomly coupled with one of the seven rating scales for each participant. More specifically, for the first excerpt and rating, participants were randomly presented with one out of 7x7 possible combinations. For the second excerpt, participants were randomly presented with one out of 6x6 possible combinations and so on. The type of combination and order of all excerpts and ratings was randomized for each participant individually. Importantly, each excerpt and rating scale was only presented or used once by each participant.

The general continuous self-report paradigm was based on Schubert et al. [53]. The study was created and carried out using the Qualtrics online platform. After providing informed consent, participants were presented with detailed instructions about their task and were familiarized with the continuous rating scales. They were then presented with the seven different combinations of musical excerpt and rating scale in individual random order. Before each excerpt, they were informed what aspect they should rate and reminded about how to do that. Excerpts played automatically and participants were instructed to rate their experiences or perceptions *continuously*, changing their rating when their experience or perception changed. For that purpose, participants were shown a Likert-type rating scale while listening to the excerpt. Each scale included five scale points ranging from ''Not at all (1)', 'A little (2)', 'Moderately (3)', 'Very (4)', to 'Extremely (5)' (with numbers in brackets referring to the scale point or the specific number key; see below for a specific example). Participants were instructed to update their response by either clicking on an option with their mouse, using the arrow keys to decrease or increase their rating, or using the number keys on their keyboard to indicate a response between 1 and 5. The lowest scale point was selected as the default when participants started a new excerpt (an example of the rating screen is provided in the S1 File).

We retained the exact time at which participants changed their rating and matched it to the time code of the musical excerpts using a JavaScript code. Musical excerpts were hosted on YouTube and embedded in the survey by hiding player controls and visuals.

## Materials

Each participant was presented with a total of seven musical excerpts. These excerpts were selected based on a pilot study (see S1 File and https://osf.io/y4dfa/), and a subset of them have been utilized in previous studies investigating the emotional impact of music [7, 52, 61]. An excerpt of two to three minutes was taken from each piece, the exact length of the excerpt depending on the phrase structure of the piece in question. Based on ratings of the pilot study, the excerpts were grouped into *sadly moving* (perceived high in sadness and feeling moved,

but low in happiness), and *joyfully moving* (perceived high in happiness and feeling moved, but low in sadness). Three excerpts were included in the sadly moving category ('Allegri', 'Olafur', and 'Oblivion'), and four in the joyfully moving category ('Band of Brothers', 'Hoppipolla', 'Vltava', and 'Explosions'). Originally, the intention was to have three excerpts each in the sadly and joyfully moving categories, as well as one 'neutral' excerpt that was perceived to the same degree as joyful and sad. However, due to a coding error in the numbering of the pilot stimuli, 'Explosions' was mistakenly identified as the neutral excerpt. As a result, no neutral excerpts were included in the main experiment. More details about this, as well as a detailed overview of each excerpt including duration is provided in the S1 File and on https:// osf.io/xgj85/.

Participants were presented with seven different scales: 'Perceived Sadness', 'Perceived Joy', 'Feeling Moved or Touched', 'Sense of Connection', 'Perceived Beauty', 'Warmth (in the chest)', and 'Chills'. Continuous rating scales targeted perceived sadness ("Right now, how sad does the music sound?"), perceived joy ("Right now, how joyful does the music sound?"), perceived beauty ("Right now, how beautiful does the music sound?"), feeling moved or touched ("Right now, how moved or touched do you feel?"), felt sense of connection ("Right now, to what degree do you feel a sense of connection?"), felt warm feeling in the chest ("Right now, to what degree do you experience a warm feeling in the chest?"), and felt chills or goosebumps ("Right now, to what degree do you experience chills (goosebumps)?"). All ratings were completed on a 5-point scale ranging from 'Not at all (1)', 'A little (2)', 'Moderately (3)', 'Very (4)', to 'Extremely (5)' (e.g., for the sadness rating: Not at all sad (1), A little sad (2), Moderately sad (3), Very sad (4), Extremely sad (5)). Note, that previous studies have typically assessed chills using a dichotomous measure probing whether participants experienced chills or not [37]. Based on Schubert et al. [53], we were interested in differentiating various intensities of chill responses.

After listening and continuously rating each excerpt, participants also completed additional measures pertaining to each of the excerpts. First, they were asked to indicate whether they had heard the piece before (with answer options: Definitely yes, probably yes, might or might not, probably not, definitely not). Then, they completed a 7-point scale asking about how much they had enjoyed the excerpt, anchored at 'Not at all (0)' and 'Very Much (6)'. We also included a self-report item asking whether participants had experienced tears or moist eyes while listening to the excerpt on the same 7-point scale. Finally, participants were presented with a dichotomous item asking about technical difficulties during playback. Upon completing the ratings for all seven excerpts, participants were asked to complete the empathic concern (Chronbach's α = .91) and fantasy (α = .85) subscales of the Interpersonal Reactivity Index (IRI; [57]), a questionnaire designed to assess interindividual differences in empathic responding. Each subscale consisted of seven items answered on a 5-point scale ranging from 'Does not describe me well' to 'Describes me very well'. Participants also completed demographic information (including gender, age, nationality, number of children, whether they had a pet, and a question about relationship status). We also included an item asking whether participants understood any of the lyrics (two of the excerpts included lyrics; *Allegri* in Latin, and *Hoppipolla* in Icelandic/Hopelandic) with the answer options 'Yes' and 'No'. Finally, we included an item to assess musical proficiency, asking whether participants play/have played any musical instruments, and if yes, for how long.

## Results

### Data preparation

An overview of the general data preparation process is provided in Fig 1. First, for each participants' timestamped ratings we created time series at 1Hz resolution (one rating per second;

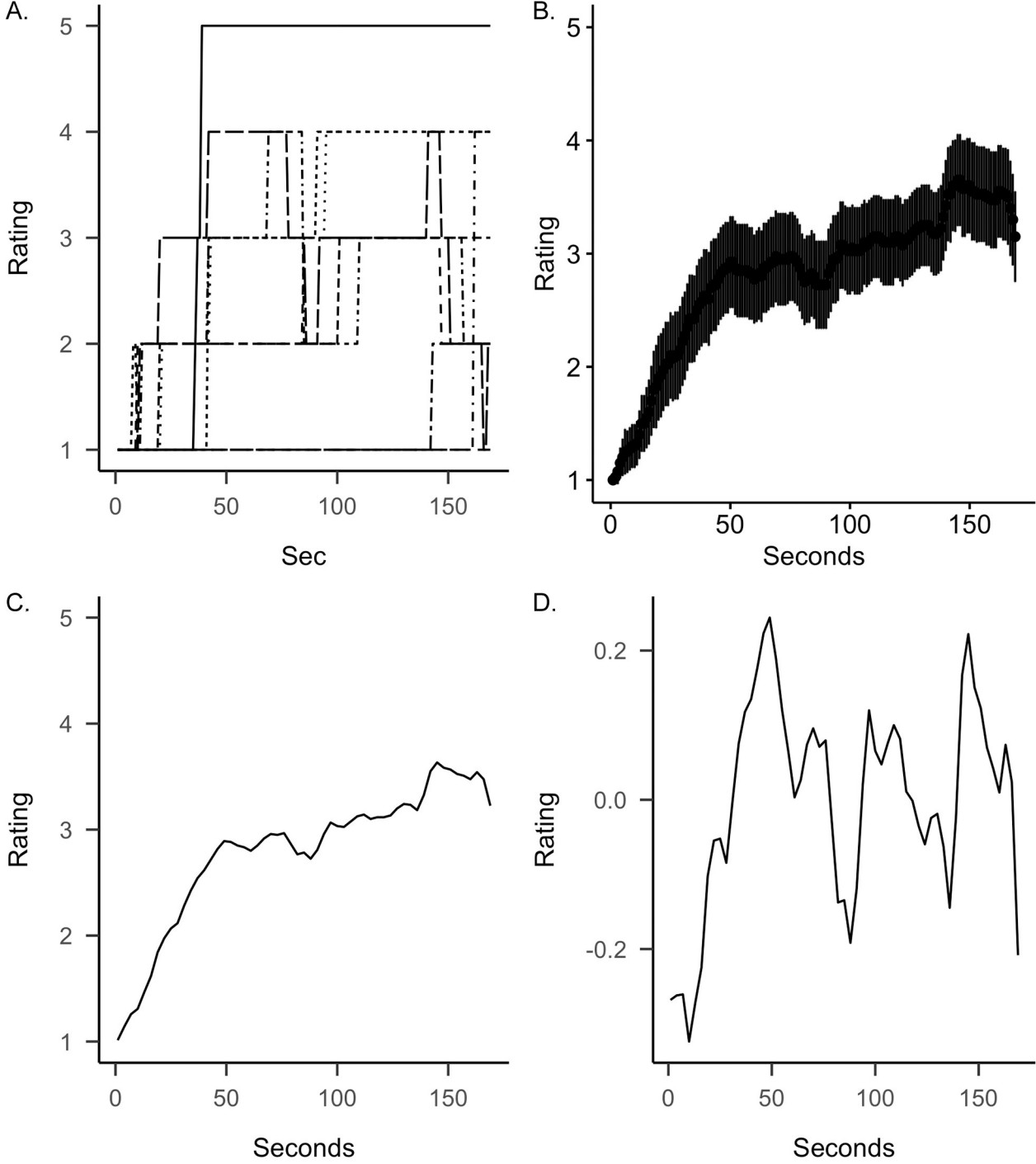

**Fig 1. Overview of the data handling process.** (A) We recorded individual time series for each scale and musical excerpt. (B) Individual time series were averaged across each scale and musical excerpt combination (error bars represent 95% CIs). (C) The time series was averaged within units of three consecutive seconds. (D) We applied a cubic spline interpolation using six sampling points in order to remove trends in the time series. The depicted example graphs represent the *feeling moved* ratings for *Explosions* as an illustration of the overall process.

Fig 1A). For example, if a participant started with a rating of 1 on a specific scale and changed their rating to a 3 at 30 s, the time series would show a rating of 1 from 0 s to 29 s, and a rating of 3 from 30 s until the second the rating was changed again (or until the end of the musical excerpt if the rating did not change). Second, we averaged the time series for each scale-excerpt combination across participants, resulting in seven averaged time series ratings for each of the seven excerpts (Fig 1B). Consistency of time series was computed with intraclass correlations, which represent a measure of inter-rater agreement (ICC; [62]). The mean ICC across all excerpt-rating combinations was .86 and all indices except the one for perceived sadness and Hoppipolla were above .60 (see S4 Table in S1 File), indicating good agreement among raters. Third, we decreased the resolution of the time scales by aggregating judgments within units of three consecutive seconds. We assumed that participants were not able to report their momentary experiences within the exact same second they appeared. Previous research has suggested that 3 seconds represent the temporal building blocks of many aspects related to human perceptual or motor abilities [63]. Therefore, we registered three seconds as an adequate time bin (Fig 1C). To test the robustness of our aggregation in the light of findings suggesting that aesthetic ratings are stable at 1s responses [64], we repeated the main cross-correlations for the 1Hz data. Effects were similar, though a bit stronger. An overview is provided in the (S8 Table in S1 File).

Fourth, we detrended the time series. Most time series are non-stationary, meaning that they show linear or higher-order increases or decreases. Comparing two non-stationary time series would inflate correlations, as they tend to cross-correlate simply due to the fact that they have trends in common. Detrending, removing the trend in time series, is one possible way to achieve the stationarity of time series data [65, 66]. Inspecting our time series, all averaged curves showed increases in ratings over time and some also decreases in ratings towards the end. More specifically, they indicated positive linear and negative quadratic trends. We detrended all time series using a cubic spline interpolation [59, 65]. Splines can be considered as an extension of polynomial regression that divides the time series into a number of k intervals, delimited by *knots*. Employing a cubic spline, a regression with three parameters (linear, quadratic, and cubic) is fit at each knot. We used five knots and therefore six intervals for all time series ([67]; Fig 1D).

We employed two additional detrending techniques: the *residual* and *difference* methods. For the residual method we removed the linear and quadratic trends from all curves by regressing them separately in a multiple regression on an index of time in seconds and its square, and saved the unstandardized residuals (Shumway and Stoffer [66]; S5 Table in S1 File). In differencing, the difference between the current observation and the previous observation is calculated for the whole time series. The direction of all main effects was similar with the three detrending methods. However, while the differencing and spline methods produced rather similar effects, the residual method showed considerably stronger effects for most measures (see S6 Table in S1 File). Note, that we originally registered to employ the residual detrending method. However, cubic splines have been considered to be superior when it comes to removing slow drifts and have been recommended to handle time series data when collecting emotional responses [59, 68]. Results of the two additional detrending methods are reported in the S1 File. We also report the results on the original un-detrended data. While these effects showed similar direction as the detrended effects, they were much stronger, which seems to be based on the underlying trends inflating and overestimating cross-correlations.

The phenomenon of feeling moved tends to happen over shorter-time intervals and is not expected to demonstrate long-term trajectories, as typically assumed for emotional experiences (in contrast to longer-lasting moods; Beedie et al. [69]). Hence for the purpose of this study detrending does not affect this significantly. Especially, the cubic spline detrending that we

employ ensures that we capture these shorter-term fluctuations while removing the long-term trajectories which are more an artefact resulting from the non-stationary nature of the emotion ratings (e.g., [70]).

After inspecting the time-series, we noticed that most time-series increased sharply for the first few seconds of the excerpts. Such early changes likely reflect the change from silence to music (rather than changes within the music) and have been called *initial orientation time*, the time it takes for continuous ratings to fall within some specific range [71]. In a previous study the median initial orientation time was observed at 8s [71]. As such sharp increases that might not reflect changes related to the music could inflate relationships across the time-series, we excluded the first three time-bins (i.e., the first nine seconds) of every excerpt. Note that this decision was not pre-registered.

### Acoustic feature extraction

Eleven acoustic features were chosen to broadly capture the loudness-related, timbral, tonal and rhythmic aspects of the musical excerpts (see Table 1 for a description of each acoustic feature). The features can typically be classified into two categories based on the duration of the analysis-window used during the feature extraction process [72, 73]. Short-term features were extracted using a 50ms window with a 50% overlap and encompassed loudness and timbral properties. While loudness was captured by the feature root-mean-square energy, timbral features primarily comprised spectral shape descriptors, namely spectral centroid, spectral spread, spectral roll-off, entropy, roughness, and flatness. In addition, spectrotemporal fluctuations were captured by spectral flux and temporal fluctuations by zero-crossing rate.

Long-term features capture context-dependent tonal and rhythmic aspects of music and are extracted using a three second window with an overlap of 67%. Tonal variation was captured by key strength or key clarity while rhythmic variation was captured by pulse clarity. All features were extracted from each excerpt using the MIRtoolbox [78]. Subsequently, spline

**Table 1. Overview and description of acoustic features extracted.**

| Acoustical Feature | Description |
| --- | --- |
| **Loudness** | |
| Root Mean Square Energy | This is the square root of the sum of squares of amplitude in a particular frame. This measures the instantaneous energy in the signal, therefore typically capturing loudness related information. |
| **Timbre** | |
| Zero crossing rate (ZCR) | Rate of the time domain zero crossings of the signal, typically indicative of either rapidly changing sonic events or sustained sounds of high frequencies. |
| Spectral centroid | Geometric centre of the frequency scale of the amplitude spectrum. This correlates with perceived brightness. |
| Spectral entropy | Relative Shannon entropy [74] indicating the presence or absence of predominant peaks in the spectrum. For example, a single sine tone has minimal entropy and white noise has maximal. |
| Spectral roll-off | The frequency below which 85% of the signal's total energy is contained. Higher Spectral roll-off may represent either higher perceived brightness and/or indicate the presence of rapidly occurring events. |
| Spectral flux | The Euclidean distance between the amplitude spectra of subsequent windows, this is a measure of the temporal change in the spectrum. Greater spectral flux is indicative of rapidly changing stimulus versus a more sustained one. |
| Spectral spread | Standard deviation of the spectrum. Noise-like signals have larger spread while single sustained tones have lower spread. |
| Spectral flatness | Ratio of the geometric mean to the arithmetic mean of the spectrum (Wiener entropy). It tends to correlate with spectral spread. |
| Roughness | An estimate of sensory dissonance [75]. |
| **Tonality** | |
| Key Clarity | Measure of the tonal clarity [76]. |
| **Rhythm** | |
| Pulse clarity | Measure of clarity of the pulse [77]. |

detrending was performed on the acoustic features, and the first 9 seconds of the excerpt were excluded from analyses as done for the emotional ratings. Finally, we created time bins of three seconds in order to match the self-report ratings. Detrending of the acoustic features was necessary to ensure comparability with the detrended self-report time series, which removed potential slow drift or long-term features. The transformation procedure was based on previous studies focusing on comparing acoustic features with neurophysiological responses [72].

## Cross-correlations

In order to compare changes in the different ratings over time, we computed cross-correlation functions (CCF) and their 95% confidence intervals. To calculate cross-correlations and estimate confidence intervals across *sadly* and *joyfully moving* excerpts, cross-correlations for each excerpt were employed in a random-effects meta-analysis using restricted maximum likelihood estimation in the *metafor* package in *R* [79]. Note that this approach was not pre-registered. Alternatively, we employed Fisher-Z transformations in order to obtain overall coefficients as specified in the pre-registration. Cross-correlations were first transformed using a Fisher-Z transformation, averaged, and finally back-transformed. Findings differed minimally and are presented in the (S7 Table in S1 File). We focused on cross-correlations at lag zero: A high cross-correlation between two scales at this lag means that both ratings change concurrently in the same direction as the music unfolds. Note, that a cross-correlation function at lag zero is the same as a zero-order Pearson correlation coefficient. For robustness, we also calculated the main analyses using Spearman correlation coefficients that are presented in the S9 Fig in S1 File. Similar to a previous study [53], we did not observe any systematic changes across other lags. An overview of the correlations between feeling moved or touched and all other continuous ratings, averaged for the different types of excerpts (after detrending employing the spline method) is presented in Fig 2. Additional results before detrending and after detrending employing the residual and difference methods are presented in S6 Table in S1 File. We also provide cross-correlations among the other variables in S9 Table in S1 File.

**Warmth, chills, and a sense of connection.** As predicted in H1, feeling moved or touched cross-correlated highly with experiencing warmth in the chest with an average of a cross-correlation function at lag 0 ($CCF_0$) of .56 [.32, .80] across all excerpts. The effect was somewhat smaller for *sadly* than for *joyfully moving* excerpts (sadly: $CCF_0$ = .50 [-.13, 1.13], joyfully: $CCF_0$ = .61 [.47, .75]), which was based on one sadly moving excerpt showing a negative effect (Oblivion: $CCF_0$ = -.17 [-.47, .13]). Similarly, feeling moved or touched cross-correlated highly with experiencing chills or goosebumps with an average of $CCF_0$ = .63 [.51, .74] across all excerpts. The effect did only differ slightly between *sadly* ($CCF_0$ = .65 [.45, .84]) and *joyfully moving* excerpts ($CCF_0$ = .60 [.45, .76]). Finally, feeling moved or touched and feeling a sense of connectedness correlated highly with an average of $CCF_0$ = .66 [.54, .76] across all excerpts. The effect was significantly stronger for *joyfully* ($CCF_0$ = .75 [.63, .86]) than *sadly moving* excerpts ($CCF_0$ = .51 [.39, .64]).

**Perceived beauty.** As predicted in H2, feeling moved or touched cross-correlated highly with perceived beauty with an average of $CCF_0$ = .60 [.49, .71] across all excerpts. The *sadly moving* excerpts ($CCF_0$ = .58 [.32, .83]) showed similar effects compared to the *joyfully moving* excerpts ($CCF_0$ = .60 [.47, .74]), though higher variation across excerpts.

The averaged (non-detrended) continuous ratings of feeling moved or touched, perceived beauty, and feeling a sense of connection for the *sadly* and *joyfully moving* excerpts are displayed in Fig 3 and the detrended (cubic spline) ratings are presented in the S6 Fig in S1 File. The (non-detrended) ratings of feeling moved or touched, chills, and warmth in the chest are presented in Fig 4 (and the detrended ratings in S7 Fig in S1 File).

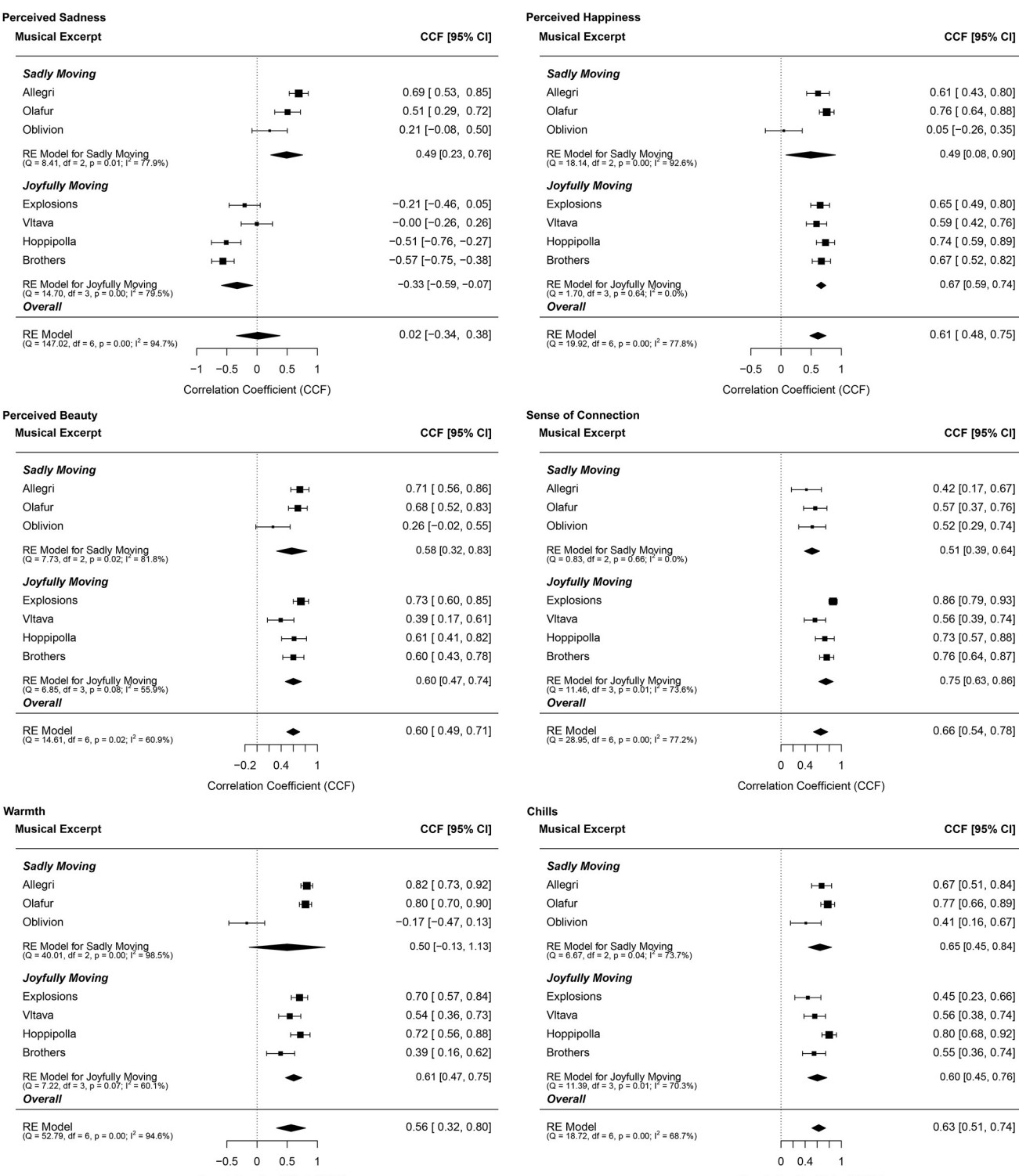

**Fig 2. Cross-correlation functions (CCF) between feeling moved and touched and the six main variables across the musical excerpts.** CCFs were calculated at lag 0 (meaning that both time series are compared at the same time) using cubic spline detrended timeseries. Sadly and joyfully moving songs are grouped separately. Overall estimates and confidence intervals are constructed employing a random-effects meta-analysis using restricted maximum likelihood estimation. Error bars represent 95% confidence intervals. RE = random effects. As heterogeneity measures we included Cochran's Q and I$^2$.

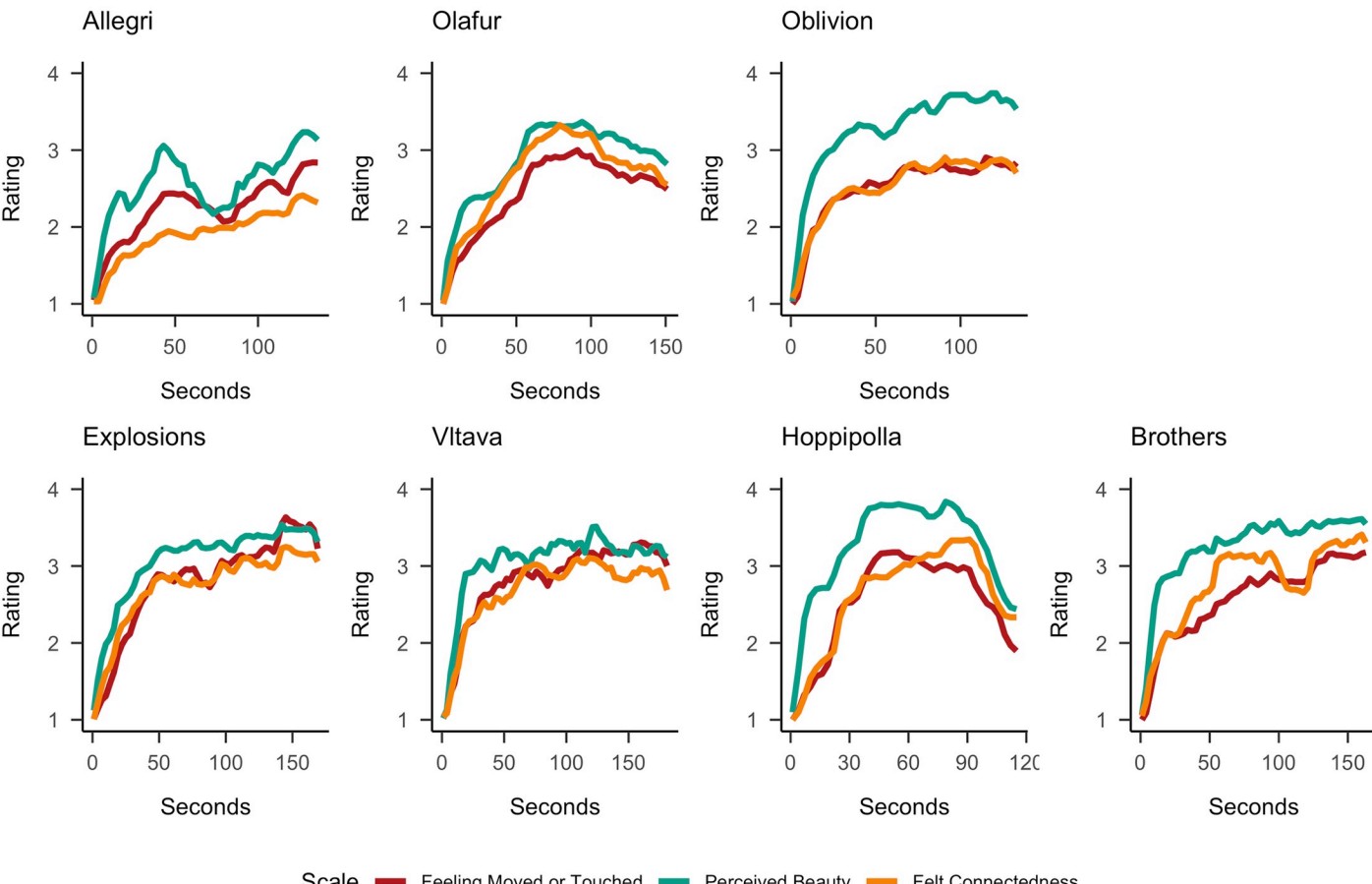

**Fig 3. The averaged non-detrended continuous ratings of *feeling moved or touched, perceived beauty*, and *feeling a sense of connection* for the sadly (Upper Row) and joyfully moving (Lower Row) music excerpts.** Ratings were provided on a 5-point scale (ranging from 1 (not at all) to 5 (extremely)) and aggregated in 3s time bins.

**Perceived sadness.**    The cross-correlation between feeling moved or touched and perceiving the excerpt as sad was $CCF_0$ = .02 [-.34, .38] across all musical excerpts. As hypothesized in H3, the effect was much stronger for the *sadly moving* excerpts ($CCF_0$ = .49 [.23, .76]) than for the *joyfully moving* excerpts ($CCF_0$ = -.33 [-.59, -.07]), which showed an effect in the opposite direction.

**Perceived joy.**    The cross-correlation between feeling moved or touched and perceiving the excerpt as joyful was $CCF_0$ = .61 [.48, .75] across all excerpts. Partly confirming predictions in H4, the effect was somewhat stronger for *joyfully moving* ($CCF_0$ = .67 [.59, .74]) than *sadly moving* excerpts ($CCF_0$ = .49 [.08, .90]), although this difference was driven by greater variation across the sadly moving excerpts. Except for one sadly moving excerpt (Oblivion: $CCF_0$ = .05 [-.26, .35]), all musical excerpts showed a strong positive cross-correlation above .59.

The averaged (non-detrended) continuous ratings of feeling moved or touched, perceived sadness, and perceived joy for the *sadly* and *joyfully moving* excerpts are displayed in Fig 5 and the detrended (cubic spline) ratings are presented in the (S8 Fig in S1 File).

## Exploratory analyses

The analyses reported in the following section were not pre-registered and/or did not test a pre-registered hypothesis, and should thus be considered exploratory.

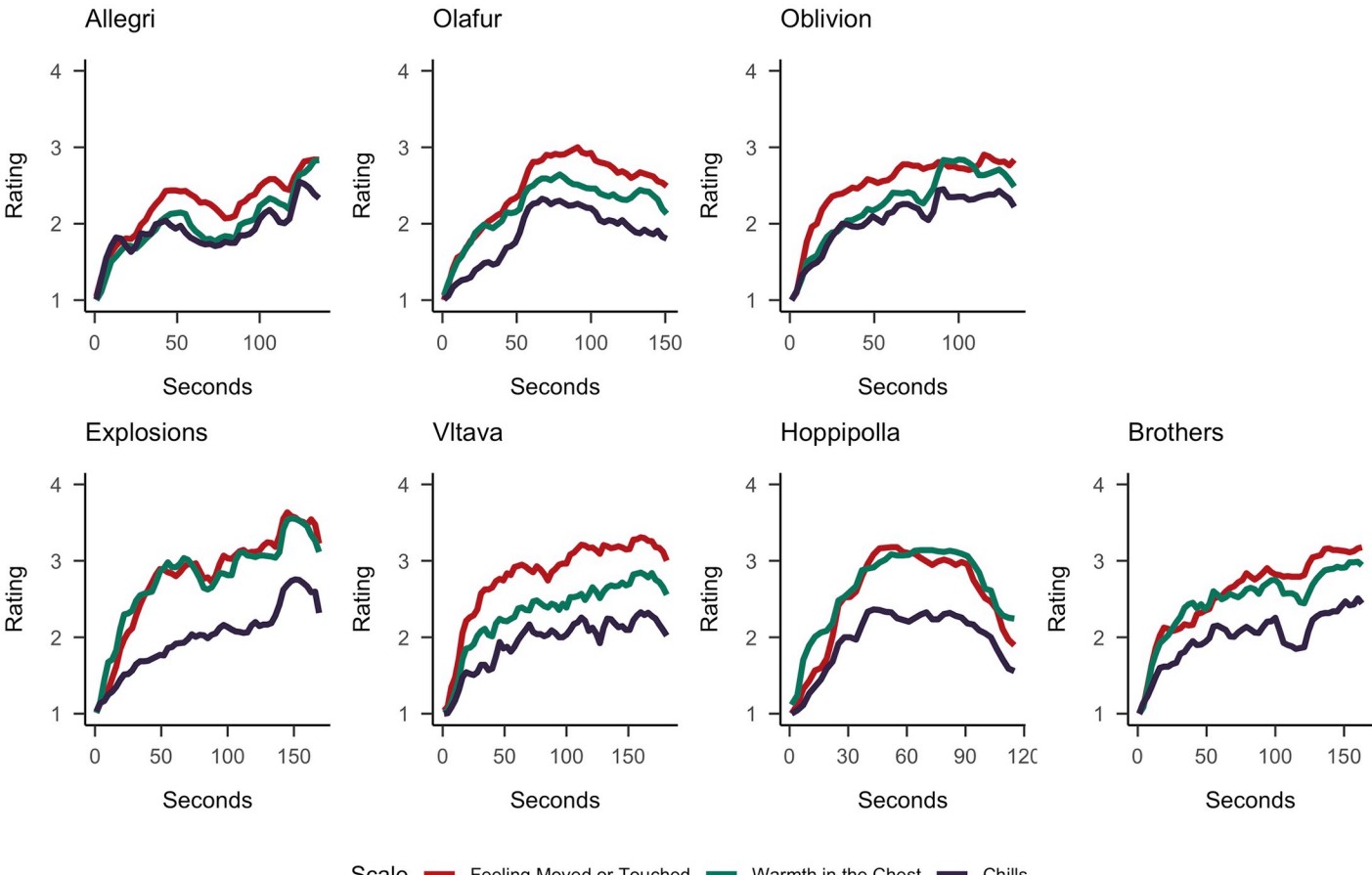

**Fig 4. The averaged (non-detrended) continuous ratings of *feeling moved or touched*, *warmth in the chest*, and *chills* for the sadly (Upper Row) and joyfully moving (Lower Row) music excerpts.** Ratings were provided on a 5-point scale (ranging from 1 (not at all) to 5 (extremely)) and aggregated in 3s time bins.

**Enjoyment, familiarity and self-reported tears.** In order to explore the relationship between *liking*, *familiarity*, and experiencing *tears or moist eyes*, liking of the musical excerpt was regressed on familiarity, tears or moist eyes and their interaction in a multilevel model. Intercepts were allowed to vary randomly according to participant and excerpt type. First, we found a main effect of familiarity. Higher familiarity with an excerpt increased the liking of that excerpt with an unstandardized coefficient of $B = -.42$ [-.53, -.32], $t(2380) = -8.05$, $p < .001$. In addition, we observed an interaction effect of familiarity and tears, $B = .08$ [.03, .12], $t(2292) = 3.33$, $p < .001$. Highly familiar excerpts were generally liked more. However, when combined with a strong response of tears, low familiarity resulted in more enjoyment than high familiarity and no or medium tear response.

**Acoustic and musical correlates of feeling moved.** Spearman correlations were calculated between the detrended continuous ratings of feeling moved or touched and the eleven acoustic features (extracted using the MIRtoolbox). Fisher's [80] Z transformation was then applied to the correlation values followed by adjusting by the factor $1 / \sqrt{(\mathrm{df} - 3)}$, where df represents the estimated number of effective degrees of freedom calculated using the approach described by Pyper and Peterman [81]. The subsequent corrected z value was then converted to a respective *p* value. This procedure is the same as that carried out in previous studies focusing on the correlation between acoustic features and continuous neural responses [72].

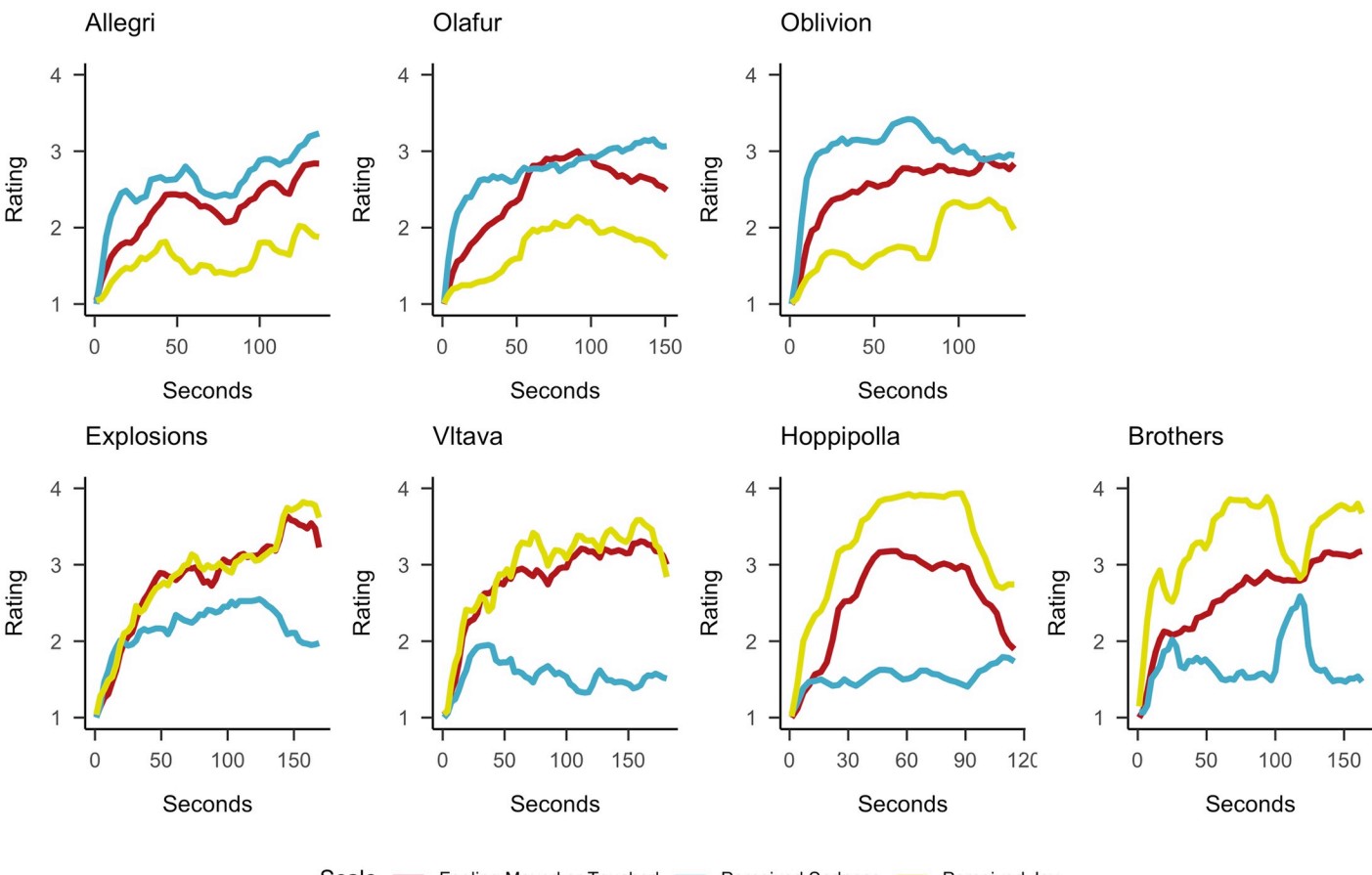

**Fig 5. The averaged (non-detrended) continuous ratings of** *feeling moved or touched*, *perceived sadness*, **and** *perceived joy* **for the sadly (Upper Row) and joyfully moving (Lower Row) music excerpts.** Ratings were provided on a 5-point scale (ranging from 1 (not at all) to 5 (extremely)) and aggregated in 3s time bins.

The results of the correlation analyses are displayed in Fig 6. *Joyfully moving* excerpts exhibited similar positive correlation profiles between feeling moved or touched and the acoustic features representing loudness (*rms*), sensory dissonance (*roughness*) and spectrotemporal variations (*flux*). In addition, *Vltava* displayed a significant positive correlation with zero-crossing rate (*zcr*) and a negative correlation with *key clarity*, while *Hoppipolla* displayed a positive correlation with spectral *rolloff*. However, among the three *sadly moving* excerpts, only *Allegri* displayed significant correlations of feeling moved or touched with zero-crossing rate (*zcr*) and spectral *entropy*, with *Oblivion* displaying a similar correlation profile.

**Correlations between mean ratings of feeling moved and trait empathy scores.** In order to explore potential individual differences in feeling moved or touched, we calculated Pearson correlations between mean moved or touched ratings (averaged across individual time-series) and the two subscales of the Interpersonal Reactivity Index: *empathic concern* and *fantasy*. We calculated an estimate across songs by running a random effects meta-analysis for each subscale. Empathic concern correlated significantly with feeling moved or touched *overall* ($r = .31$ [.22, .41]), as well as by both *sadly moving* ($r = .26$ [.12, .41]) and *joyfully moving* excerpts ($r = .35$ [.21, .48]). Fantasy showed a smaller overall effect ($r = .11$ [.01, .21]), as it did not correlate significantly with feeling moved or touched by *sadly moving* excerpts ($r = -.01$ [-.16, .15]), and only modestly with feeling moved or touched by *joyfully moving* excerpts ($r =$

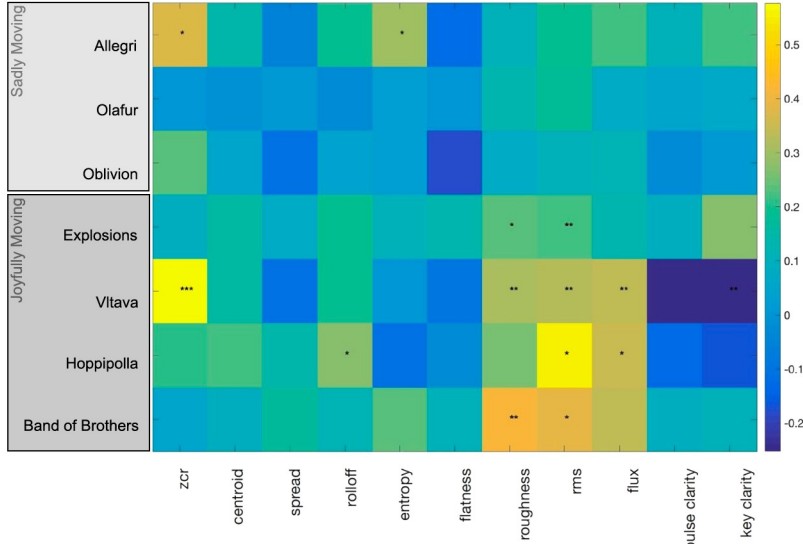

**Fig 6. A visualization of the spearman correlations between continuous ratings of _feeling moved or touched_, and the acoustic and musical features of the music excerpts.** Color gradients represent the strength of the correlation coefficient. $^{*}p < .05$, $^{**}p < .01$, $^{***}p < .001$.

.20 [.07, .34]). We also explored the correlations between peak feeling moved (instead of mean ratings) and trait empathy scores (see 2.3 Section in S1 File). Findings were nearly identical.

Finally, since the mean ratings of feeling moved or touched revealed significant correlations with trait empathy, we decided to explore individual differences in continuous rating patterns in more detail. It may be that the higher mean values of feeling moved exhibited by the high-empathy participants either reflect higher overall levels of feeling moved, or more pronounced peaks or variations in the continuous ratings. In order to identify potential groups of people with similar rating trajectories, we performed a clustering analysis. The detailed results can be found in the 2.4 Section in S1 File. Overall, we found evidence that higher trait empathic concern was associated with higher peaks of continuous feeling moved ratings, though only for two musical excerpts: Hoppipolla and Band of Brothers.

## Discussion

The purpose of this study was to explore continuous ratings of feeling moved or touched by music, and how the time course of feeling moved might covary with specific appraisals, emotional perceptions, and bodily sensations. In addition, we explored the association of continuous ratings of feeling moved with acoustical features extracted from the musical excerpts. Building on previous theorizing and studies on feeling moved (e.g., [6, 22]), we hypothesized that ratings of feeling moved would cross-correlate with feeling a sense of connection, experiencing chills, and experiencing a warm feeling in the chest (H1). Furthermore, because previous studies associated feeling moved with the perceived beauty of music [7], we hypothesized that feeling moved or touched would cross-correlate with ratings of perceived beauty (H2). Finally, we hypothesized that feeling moved or touched would cross-correlate with perceived sadness in the case of sadly moving excerpts (and less so for happily moving excerpts; H3), while it would cross-correlate with perceived happiness for joyfully moving excerpts (and less so for sadly moving excerpts; H4).

Most of our pre-registered hypotheses were supported by our findings. Figs 4–6 show that the time course of feeling moved largely coincided with that of sense of connection, chills, warmth in the chest, perceived beauty and perceived joy across all musical excerpts, and with perceived sadness for sadly moving excerpts. The preregistered and additional cross-correlation analyses confirmed these impressions. A further inspection of the time courses of these ratings shows that they typically increase along the course of the excerpt, with a rather strong initial rise within the first 30 seconds, resulting in a positive linear trends in all ratings and excerpts. Notably, we cannot completely disentangle whether such a rise was caused by actual sudden changes in experiences or by a general need to change the rating because it always started at the lowest score. Furthermore, some excerpts show only monotonic increases, while others had peak ratings before the end and then declining ratings, and some more than one peak. These different time courses are not necessarily reflective of the dynamics of whole musical pieces, as we only used specific excerpts. They are similar in shape to the ones found for moving videos [53].

Continuous ratings of feeling moved or touched had one of the highest cross-correlations overall with ratings of "feeling a sense of connection", suggesting that appraisals or experiences of closeness or affiliation are associated with experiences of feeling moved also in the case of music. In contrast to a previous study in which people rated the perceived closeness of the people appearing in the video clips [53], we chose to operationalize closeness more broadly as "feeling a sense of connection". The exact nature of the connection experienced was intentionally left open in order to accommodate for the "floating intentionality" inherent in music [82]. Thus, it is possible that this 'sense of connection' may have been construed as existing between the listener and another agent (the music, the performer, other listeners, or perhaps humanity in general), or between real or virtual agents perceived in the music (cf. [8]). In terms of the intensification of communal sharing relations central to kama muta theory, these 'connections' could be conceptualized in terms of first- or second-person empathy and compassion (e.g., empathizing with the music or experiencing the music as empathizing with oneself; see [9, 83, 84]), or in terms of affiliative intentions perceived in the music [8]. Despite the range of possible relationships and interpretations, our results demonstrate that participants rated "feeling a sense of connection" consistently, suggesting a shared understanding of the concept and the cues contributing to it.

Our findings also confirmed that ratings of feeling moved or touched cross-correlated with self-reported chills or goosebumps and experiencing a warm feeling in the chest in response to music, replicating previous findings on the association of these variables (e.g., [5, 22, 43, 45]). These patterns of cross-correlations were similar across both sadly and joyfully moving excerpts. This indicates that feeling moved by music is associated with a similar pattern of physiological or bodily sensations as feeling moved by videos depicting social scenarios (and experiences of feeling moved in general; [5]), and that this pattern does not depend on the perceived sadness or joyfulness of the music.

Interestingly, while Konečni [18, 19] predicts chills to be a more frequent response to music than feeling moved, we found high cross-correlations between the two, suggesting that they largely coincided in occurrence. Furthermore, in line with a previous study showing that movingness mediates the relationship between perceived sadness and beauty in music [7], we found positive cross-correlations between ratings of feeling moved or touched and perceived beauty. Interestingly, these cross-correlations were consistent across both sadly and joyfully moving excerpts. Previous research has associated perceived beauty with perceived sadness and only to a lower degree with perceived happiness ($r = .59$ vs. $r = .16$; Eerola and Vuoskoski [85]), while in the current study we found overall weaker associations between perceived sadness and beauty than between perceived joy and beauty ($r = .12$ vs. $r = .54$).

Supporting previous findings [86], *moving* and *touching* music was also perceived as beautiful. This bears the question whether feeling moved by music triggers a perception of beauty or vice versa. Is all music experienced as beautiful automatically experienced as moving or is moving music automatically perceived as beautiful? Further research is needed to investigate this possibility.

Finally, we had hypothesized that feeling moved or touched would cross-correlate with perceived joy for joyfully moving excerpts but less so for sadly moving excerpts, and with perceived sadness for sadly moving excerpts but less so for joyfully moving excerpts. While feeling moved cross-correlated with perceived joy, but not sadness for joyfully moving music, the picture regarding sadly moving music turned out to be somewhat more complex: When the overall emotional tone of the music was sad rather than joyful, feeling moved or touched cross-correlated with *both* perceived sadness and perceived joy (except for the excerpt "Oblivion"). The finding that ratings of feeling moved or touched cross-correlated positively with perceived joy across both joyfully and sadly moving excerpts, supports previous theories suggesting that the overall tonality of the experience of feeling moved is positive [6]. This is in fact reflected in most perspectives on feeling moved (cf. [87]).

The finding that feeling moved or touched cross-correlated positively with perceived sadness for sadly moving excerpts can be interpreted from two different angles. First, it could mean that, although predominantly positive, feeling moved constitutes a mixed state including both negative and positive affect, as suggested by one specific theory [21]. Second, this could mean that feeling moved is experienced as predominantly positive but can co-occur with other negative emotions such as sadness, as suggested by kama muta theory [12, 15]. The observation that feeling moved or touched cross-correlated negatively with perceived sadness for joyfully moving excerpts provides some evidence for the latter explanation. However, it should be noted that we assessed perceived and not experienced sadness and joy, which means that our findings can only be interpreted as indirect evidence. In sum, the main findings support the idea that feeling moved is perceived as predominantly positive—also in the context of music.

Not only did we obtain strong cross-correlations between feeling moved or touched and the other variables (except for perceived sadness in the joyfully moving and overall context), we also observed strong associations among perceived joy, a sense of connection, and reported chills and warmth. These findings support kama muta theory [15] and the theory by Menninghaus et al. [21]. Kama muta theory argues that the underlying emotional construct represents the co-occurrence of sudden intensifications of communal sharing relationships, experiences of tears, chills, and/or warmth in the chest, motivations to act on one's communal sharing relations, positive valence, and labels such as *moved* or *touched* [15]. While the kama muta framework has been supported for short film clips [5], this is the first systematic evidence for the configuration of kama muta in response to music (see [31]). Notably, we did not assess the motivational component of kama muta as it is currently unclear how this might relate to the reception of music.

A further theory conceptualizing experiences of feeling moved is core values theory [88]. Core values are defined as those that are of central importance to a social group. Connection is presumably of central importance to most groups, but so are other values such as achievement [89]. Our findings are thus compatible with the core values theory of feeling moved but do not provide evidence that values other than connection contribute to being moved by music. Our paradigm could be used to assess perceived skill or virtuosity of the musicians and study its cross-correlation with feeling moved. According to core values theory, there should be a positive cross-correlation to the extent that being a skillful musician is seen as an admirable value in that cultural context.

Finally, aesthetic trinity theory [18, 19] conceptualizes feeling moved as a highly idiosyncratic response to the sublime, and chills as a more frequent response to music. Our results

lend partial support to that theory, as 'sense of connection' can refer to connection with the sublime. However, given previous work on music and the sense of connection it conveys (e.g., [90]), we suggest that it mainly refers to social connection perceived between the self and the music or between agents perceived in the music. Furthermore, we found high inter-rater agreements for all scales. The ICC for sense of connection was .84 or higher for all musical excerpts studied, suggesting that there was high interpersonal agreement on which passages conveyed more connection than others. This suggests that the experience of feeling moved by music can be reliably evoked by the same musical passages across participants. It is worth noting that we had chosen these excerpts to be moving. Other musical pieces can be expected to result in lower inter-rater agreements, and thus more idiosyncratic experiences of feeling moved. Still, our findings suggest that feeling moved can be reliably evoked by particular musical features, independent of individual recollections and associations.

## Acoustic correlates of feeling moved

Our exploratory analysis of the acoustic and musical correlates of continuous ratings of feeling moved or touched revealed a pattern of acoustic correlates for the joyfully moving excerpts, while no consistent patterns were observed for the sadly moving excerpts. Continuous ratings of feeling moved or touched correlated positively and significantly with loudness (rms energy) in the case of all joyfully moving excerpts. For three of the four joyful excerpts, feeling moved or touched also correlated with spectral roughness (or sensory dissonance). In the case of the excerpts used in the present study, higher values in this feature likely correspond to increased complexity of the spectral content, reflecting multiple instruments sounding together. For two of the joyful excerpts, Hoppipolla and Vltava, feeling moved or touched also correlated positively with spectral flux. Previous work investigating the acoustic correlates of felt and perceived emotions in music has associated loudness, spectral flux and roughness with the arousal dimension of affect (e.g., [91, 92]). It may be that, at least in the case of joyfully moving music, arousal-related musical features contribute to feeling moved by increasing felt arousal.

However, in the case of the sadly moving excerpts, feeling moved or touched only correlated significantly with zero-crossing rate and spectral entropy, and only in the case of one excerpt—*Allegri*. Zero-crossing rate generally reflects rapid fluctuations in the temporal domain and has been broadly used as a feature to detect the presence of vocals or voice-like sounds and voice activity [93, 94]. Specifically, sustained sounds tend to have greater zero-crossing rate than percussive sounds. *Allegri* is characterized mainly by sustained vocals, while *Vltava* and *Oblivion* are dominated by string-instruments that carry the melody in a continuous fashion.

Overall, it is possible that arousal is less important for feeling moved evoked by sad (compared to joyful) music, but future studies should specifically investigate whether feeling moved by sad versus joyful music is associated with different levels of psychophysiological arousal. On the other hand, timbre or choice of instruments might be more important for feeling moved by sad music [95]. Moreover, several studies have emphasized the importance of lyrics especially in sad music contributing to feeling moved [96]. This warrants further study.

Importantly, the current findings are based on a small number of musical excerpts so caution should be applied when interpreting these findings, especially regarding the inconsistent results for sadly moving excerpts. In addition, since continuous ratings of feeling moved cross-correlated strongly with perceived joy and perceived beauty, it is possible that the observed acoustic features are not specific to feeling moved, but rather reflect co-occurrences with other evaluations or experiences. Altogether, we observed correlational evidence that can guide

future studies in experimentally investigating different musical aspects and measuring ratings of feeling moved (e.g., [39]).

## Trait empathy and feeling moved by music

Finally, we explored individual differences in feeling moved or touched by music. Empathic concern, the tendency to respond sympathetically to others in need [57], correlated positively with mean feeling moved or touched ratings in the case of both joyfully and sadly moving excerpts. This finding corroborates and extends previous work that has associated empathic concern with feeling moved or kama muta in response to videos, sad music, and other stimuli [7, 52, 58]. In an auxiliary analysis, we also found that empathic concern was associated with higher peaks of feeling moved—though only for two joyfully moving musical pieces, which should be interpreted with caution.

Recent studies found that empathic concern was the only subscale of trait empathy (as assessed by the IRI; Davis [57]) that was consistently associated with feeling moved (as well as reported tears, chills, and warmth) in response to videos and written narratives [5, 58]. The present findings suggest that a similar relationship exists for the case of music. Zickfeld et al. [58] argued that state empathic concern could be considered as a special case of intensifications of communal sharing and thereby a type of feeling moved or kama muta. Such a relationship could explain the cross-correlation of feeling moved with perceived sadness. Perceiving someone in need can be evaluated and experienced as negative or also sad, whereas sympathizing with the needy person can be considered as intensifying one's communal bonds, thereby inducing feeling moved [58]. How this relates to the context of music is less clear, although recent work suggests that listeners may experience feelings of compassion when listening to sad music [84]. The current results suggest an important role of empathic processes in emotional responses to moving music (see [83, 84]), and are in line with the view of musical experience as inherently social (e.g., [36]). Previous studies have argued that empathic engagement with music could take the form of resonating with the expressions and imagined experiences of the performer or the composer (e.g., [27]), or identifying with an imagined narrative or a virtual persona represented by the music [97].

Fantasy, the ability to transpose oneself into fictional situations or stories, another facet of trait empathy that we assessed [57], has previously been associated with feeling moved by sad music [7, 52]. However, in the present study, fantasy only showed stronger correlations with ratings of feeling moved or touched in response to joyful music, but less so in response to sad music. One difference between the previous studies versus the current study is that we averaged all feeling moved ratings across the whole time series, whereas previous studies used a summary rating of feeling moved after listening to the piece. Thus, it may be that individuals with higher levels of fantasy are not more moved by sad music throughout, but rather respond more strongly to highly moving passages, which may be better reflected in their own summary judgement rather than in averaging across the whole piece. Furthermore, they may remember these peak moments better than people lower in fantasy, or reconstruct the piece as more moving when asked about it in retrospect, thus providing a higher rating. By systematically comparing continuous with summary ratings, future research can directly test these propositions and quantify the contribution of traits, musical features and memory processes to ratings of feeling moved or touched.

## Limitations and future directions

An obvious limitation of the present study is that only seven music excerpts were used as stimuli. However, this aspect of the experiment design was closely tied to the constraints of the

continuous rating paradigm. This limitation is particularly relevant for the analysis of the acoustic and musical features contributing to feeling moved, since the pattern of results may be specific to the particular music excerpts. It is likely that the acoustic and musical characteristics associated with feeling moved vary somewhat from piece to piece, and thus our findings reflect only a part of the entire picture. Moreover, the musical features that were explored in the present study do not represent an exhaustive selection of possible musical features, and thus there may be other, additional musical or acoustic features that contribute to feeling moved or touched. It is also likely that the relationship between musical features and perceptual ratings is non-linear, limiting the explanatory power of linear analysis methods such as those used in the present study. Similarly, we observed higher heterogeneity in effects for the *sadly moving* excerpts in comparison to the *happily moving* excerpts. This was mostly driven by *Oblivion* that typically showed smaller effects than the other two *sadly moving* excerpts, but sometimes even the opposite as for example for the relationship between feeling moved or touched and warmth in the chest. It is unclear whether that particular excerpt represents a more prototypical or more atypical version of the category of *sadly moving* music. Future studies employing a larger stimulus pool could remedy this shortcoming.

The generalizability of the current findings hinges not only on the limited stimulus pool, but also on the employed sample. Amazon MTurk participants have been repeatedly criticized for providing low-quality data because they do not invest sufficient effort, are non-naive, and less trustworthy (e.g., [98]). However, the evidence for these claims seems to be rather mixed and dependent on specific situations [99]. Hauser et al. [99] highlight that high-quality data can be collected from MTurk, given specific research designs and an increased focus on attention checks. In the present study, we employed several attention checks and tried to make sure that participants understood the main task in order to avoid the possible pitfalls associated with MTurk participants. The fact that we replicated the main findings from a previous study using a similar paradigm [53] also speaks to the validity of our results. Nevertheless, we acknowledge that conclusions beyond the present sample and also regarding possible cross-cultural differences in being moved responses (see [5]) are necessarily limited.

However, our findings are an important first step towards understanding the contributing musical features, as well as the extent to which these may vary between sadly and joyfully moving music. Future studies could investigate the contributing features in a larger and more varied set of music examples and more diverse cross-cultural populations. A possibility for targeting such larger stimuli sets could be to instruct participants to self-select moving music (e.g., [43, 100]). While such a design minimizes the control over the stimulus material, it would be interesting to see whether the current findings replicate when employing such a technique. Furthermore, other types of analyses and experiments might shed more light on the musical cues and appraisals that are associated with feeling moved. For example, is feeling moved dependent on experiencing music as conveying prosocial intentions, or perceiving musical events in terms of prosocial interactions between agents (cf. [8])?

Another limitation is related to the continuous rating paradigm itself, where the participants are asked to continuously monitor their subjective experience and adjust their ratings accordingly. It may be that the additional cognitive load of monitoring one's internal states might detract from the intensity of the experience itself. We tried to mitigate the potential effects of this possibility by having participants only use one continuous scale per excerpt, and by using a Likert-type scale with 5 steps rather than a visual analogue slider, for example, making it less likely for participants to have to constantly adjust their rating. A complementary approach for future investigations could be to measure psychophysiological responses with and without continuous ratings, for example, in order to obtain an additional index of emotional arousal. This could also help with identifying peaks in physiological arousal, and

whether these coincide with self-reported emotional peaks. Based on our design we were not able to assess intra-individual processes, as participants never rated more than one scale for the same musical piece. All presented comparisons were made on an inter-individual level. While it is intriguing to see the amount of convergence for averaged ratings of different individuals for the same musical excerpt, this approach might have overestimated correlations among the different ratings, as within error variance remains undetected.

We also included two musical excerpts that featured vocals. While none of the participants were proficient in the depicted languages, the human voice is known to transport specific emotions as well as empathy (e.g., [101]). Nevertheless, the two vocal pieces showed similar results as the instrumental excerpts. In addition, we assessed perceived joy and sadness in the music, and not experienced joy and sadness induced by the music. Previous studies [7] and our pilot study suggested that ratings of perceived and experienced emotions in music correlate highly. Nonetheless, it should be emphasized that our measures only allow for indirect inferences about participants' experiences of joy and sadness.

Finally, in contrast to previous studies employing dichotomous ratings (e.g., [43]) we measured subjective chills using a continuous scale (based on Schubert et al. [53]). This was done in order to assess different intensities of chills or goosebumps. Importantly, we might therefore consider responses that would not be classified as chills responses with dichotomous rating paradigms. This is also obvious in the rather long average duration of chills ratings above the lowest scale point, suggesting that we not only covered peak-states but also lower intensities of subjective chills. Overall, we replicated associations between chills and different variables such as feeling moved, pointing to the validity of this measure. Notably, previous studies also differ in the reported average length of chills responses [43, 102], which is probably related to different methodologies and the exact operationalization of the phenomenon (i.e., as *chills*, *goosebumps*, or *thrills*). It is possible that for example *thrills* refer to a higher intensity response than *goosebumps* or that several short *chill* episodes occurring in quick succession are perceived as one longer response. A more standardized definition of the phenomenon of chills (see [44]) would be helpful in comparing different findings and evaluating the degree to which they investigate the same phenomenon.

## Conclusion

In sum, the findings of this study demonstrate that musically evoked experiences of feeling moved are associated with a similar pattern of appraisals, physiological sensations, and trait correlations as feeling moved by videos depicting social scenarios. The observed pattern of components is consistent with the predictions of different theories on feeling moved, specifically kama muta theory. Feeling moved or touched by both sadly and joyfully moving music was associated with experiencing a sense of connection and perceiving joy in the music, while perceived sadness was associated with feeling moved or touched only in the case of sadly moving music. Acoustic features related to arousal contributed to feeling moved only in the case of joyfully moving music. Finally, trait empathic concern was positively associated with feeling moved or touched by music. These findings support the role of social cognitive and empathic process in music listening, and highlight the social-relational aspects of feeling moved or touched by music.

## Supporting information

**S1 File. Supplementary material.** Supplementary information, including S1–S9 Tables and S1–S9 Figs.
(DOCX)

## Author Contributions

**Conceptualization:** Jonna K. Vuoskoski, Janis H. Zickfeld, Beate Seibt.

**Data curation:** Jonna K. Vuoskoski, Janis H. Zickfeld.

**Formal analysis:** Jonna K. Vuoskoski, Janis H. Zickfeld, Vinoo Alluri, Vishnu Moorthigari.

**Writing – original draft:** Jonna K. Vuoskoski, Janis H. Zickfeld.

**Writing – review & editing:** Jonna K. Vuoskoski, Janis H. Zickfeld, Vinoo Alluri, Beate Seibt.

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
