## [Decision Letter · Decision Letter 0]

25 Nov 2020

PONE-D-20-21715

Feeling moved by music: Investigating continuous ratings and acoustic correlates

PLOS ONE

Dear Dr. Vuoskoski, dear Jonna, 

Thank you for submitting your manuscript to PLOS ONE. After careful consideration, we feel that it has merit but does not fully meet PLOS ONE’s publication criteria as it currently stands. Therefore, we invite you to submit a revised version of the manuscript that addresses the points raised during the review process.

We look forward to receiving your revised manuscript.

Kind regards,

Stefan Koelsch

Academic Editor

PLOS ONE

Journal Requirements:

2.) Your ethics statement should only appear in the Methods section of your manuscript. If your ethics statement is written in any section besides the Methods, please move it to the Methods section and delete it from any other section. Please ensure that your ethics statement is included in your manuscript, as the ethics statement entered into the online submission form will not be published alongside your manuscript.

3.) Please include captions for your Supporting Information files at the end of your manuscript, and update any in-text citations to match accordingly. Please see our Supporting Information guidelines for more information: http://journals.plos.org/plosone/s/supporting-information

Reviewers' comments:

Reviewer's Responses to Questions

**Comments to the Author**

1. Is the manuscript technically sound, and do the data support the conclusions?

Reviewer #1: Partly

Reviewer #2: Yes

Reviewer #3: Partly

Reviewer #4: Yes

2. Has the statistical analysis been performed appropriately and rigorously? 

Reviewer #1: I Don't Know

Reviewer #2: Yes

Reviewer #3: Yes

Reviewer #4: Yes

3. Have the authors made all data underlying the findings in their manuscript fully available?

Reviewer #1: No

Reviewer #2: Yes

Reviewer #3: Yes

Reviewer #4: Yes

4. Is the manuscript presented in an intelligible fashion and written in standard English?

Reviewer #1: Yes

Reviewer #2: No

Reviewer #3: Yes

Reviewer #4: Yes

5. Review Comments to the Author

Reviewer #1: The paper describes one on-line study in which listeners heard 7 pieces of music and provided continuous (or at least, frequently sampled) responses on 7 different scales (sadness, happiness, feeling moved, sense of connection, beauty, warmth, and chills). In general, the ratings across scaled varied similarly over time, however for the songs that were considered “sadly moving” (meaning previously considered to be moving in a sad way), ratings of feeling moved were more correlated with ratings of sadness than ratings of happiness. Conversely, for the songs considered “joyfully moving,” ratings of feeling moved correlated more with ratings of happiness than of sadness. They also showed some correlations between acoustic features for some songs and some rating scales, as well as a correlation between “feeling moved” ratings and subject’s trait empathic concern.

The continuous ratings are an interesting way of capturing the real time musical listening experience and it is interesting to see that that distinct rating scales with separate groups of listeners have such similar time courses. In my view the biggest drawback to the paper is an overall lack of clarity in the background or theoretical justification, as well as in the methods. First, the introduction is about twice as long as it should be, and from the start it could be much clearer what questions are being asked and how they fit into what is already known. The literature review is somewhat vague on details, at least for some studies (see specific notes below). I was completely lost on the distinction between “feeling moved” and “being moved” and the “kama muta framework.” What are the differences between these constructs and do they matter for the current study? What is the relationship between “feeling moved” and feeling chills? How is this study novel given that similar approaches have been used to understand acoustic correlates of chills? The entire theoretical background could be shorter, simpler, and clearer.

My second main concern concerns the clarity of method and how the researchers assessed the validity of the scales. It’s clear that subjects were asked to use 7 different rating scales across 7 pieces, but the paper does not say what they were told about these scales. I’m particularly worried that subjects don’t really have distinct concepts of some of these scales (like beauty vs. feeling moved vs. feeling connected)—how do we know they aren’t just treating multiple scales as being about something like general emotionality or aesthetic intensity or something like that? It helps that sadly moving and joyfully moving pieces showed the distinct patterns that they did, but I’m wondering about the extent to which listeners really understood the rating scales they were using while doing the task. If the same listener had used the same scale on 7 different pieces, then this would at least allow them to compare how that listener tends to use the scale (do they tend to stay high or low on the scale, do they use a wide or narrow range, etc). But the current design doesn’t even allow for within-subject normalization within a given rating scale. It was also unclear to me what happened on a given trial. The manuscript states “participants were presented with 5-point Likert-type scales while listening to the excerpts, and asked to continuously change their ratings while the music progressed.” How did the researchers ensure that they did this? Were they probed every second? Did they have a slider? Please clarify.

p. 3, first paragraph: I do not understand the distinction that is being made between “culturally and socially constructed” emotional interpretations of music and an “embodied, empathic basis” for musical emotion. Can’t it be both?

p. 4: the claim is that music can express “social attitudes” but the study described used forced choice responses. It isn’t clear that subjects would spontaneously use terms such as “disdainful” to describe a musical performance, and it’s quite possible they are responding to more general emotional categories in this task.

p. 4: I think the authors meant to cite a different paper than Mehr & Spelke (2018)-- perhaps the Mehr (2018) Current Biology paper?

p. 9-10: What is meant by “the opposite direction” re: chills? Some studies find that chills are frequent and others find that they are infrequent? The summary on chills is very unclear.

p. 15: why were American subjects recruited for this study?

p. 16: The osf link just goes to a spreadsheet and some R code. Are the stimuli or a stimulus available somewhere?

p. 20: the cross-correlations for ratings are over means across all subjects who used the rating scale, correct? No individual subject time series are being used?

p. 24: it would be helpful to have a little more information about the acoustic measures that were used. More is said in the discussion but it would be good to also have it in the results section (like, what does zero-crossing rate mean).

Reviewer #2: This is a very interesting paper by very esteemed scholars. It proposes a very strong methodology for rating emotional reactions “while” listening, and not “after” listening. Also the search for underlying psychological mechanisms is interesting, especially the time specificity of the reactions. The strength of the paper, however, is also its mean weakness. The methodology is very strong but is too technical to be understood by common readers. A major effort should be made therefore to make the paper more readable and understandable by providing more intuitive descriptions of the technicalities, without giving in on depth of elaboration. As I consider this paper to be a very valuable contribution, I did a very demanding review in order to be as critical-constructive as possible. It means also that I would like to see this paper to be published on condition that my remarks and comments are addressed appropriately.

Note: I have commented on the paper in the order of reading, to show what readers may experience as problems why reading. Many of my remarks are addressed and even solved at later passages in the text, but for the noninforemed listerern this comes too late. In that case, it could make sense to move these passages to earlier places in the text to make reading more easy. I therefore decided not de delete these comments. I did also a first reading of the paper without reading the supplementary materials to have an idea of the readability of the paper. This was a hard but very instructive exercise.

General remarks

• Language use is OK.

• Authors show sufficient knowledge of the field.

• The reference list is quite exhaustive though some reference to real-time listening and musical emotions could be added.

• The methodology, though sound, is not well described. It could be explained more clearly by providing a more schematic overview of the major research questions. This holds in particular for the way how the continuous rating is assessed. Even after reading the whole paper this is not clear as a take-home message.

• The readability of the paper is rather low, given the many technical terms without clear description. At times the reader has the impression that this is a method-driven paper rather than a paper about interesting and clear contents. There is, as such, a kind of imbalance between empirical facts and data (measurements) and clear intuitions and take-home messages. There is a lot of technical stuff, which is not sufficiently explained, even for readers with a lot of knowledge of the field. The authors should keep in mind that to understand the paper readers should be a kind of combination of acousticians, mathematicians, engineers, statisticians, and musicians. Given the interdisciplinary team of authors, this is no problem for the authors, but it is very hard to find readers who combine these competences. Therefore, I emphasize repeatedly the need of clear and intuitive descriptions of the intuitions behind the measurements so as to reach a broader readership.

• The clarity of the paper could be better. At times it is difficult to understand the methodology with the needed description being provided “after” the facts. Some restructuring is necessary to help the reader to understand the design and the assessment.

• Some very technical procedures are not explained. Provide at least some intuitive descriptions (e.g. the Interpersonal Reactivity Index, detrended correspondence analysis, cubic spline detrenching, Ward’s minimum variance method , Calinski-Harabasz index, Mann-Whitney U test, Kruskal-Wallis-H test, etc.) as not all readers have the needed statistical background to understand the methods used. It is perhaps a suggestion to insert a box with some technical terms and their explanation to make the paper more readable.

• The time-series must be explained more clearly, given their importance throughout the paper.

• The figures are very interesting and insightful. In order to be more easily interpretable, however, they must be explained better, either in the main text or in the figure captions. This holds especially for the abbreviations and the units used. A figure, together with its figure caption, must be understandable at least to some extent without reading the main text.

• Some concepts are introduced without clear description (e.g. self scale, zero-crossing rate, entropy): some intuitive descriptions should be provided to guide those readers who are not familiar with the field.

The concepts of kama muta needs a clear definition.

• The major question how the continuous rating was assessed is not explained sufficiently clear. It seems hard to imagine how listeners can fill in a rating for a scale every second or something like this. Even after reading the whole paper, this procedure is not clear. It should be explained much more clearly, given the importance of the procedure for the whole paper and its interest for the broader readership which I consider to be very great.

• There is an inflation of statistical procedures and techniques which make reading very hard and which do not always add much to the main findings. There is a risk of method-driven analysis with the danger that the reader does no longer see the forest through the trees.

Note: I have listed my comments in order of appearance while reading the paper. Some questions or remarks are addressed appropriately, but they come too late, which makes it difficult for the reader to understand the methodology. Some restructuring and moving of some sections to earlier places in the paper seems necessary therefore. Now, as a reader, we have the impression to be informed sometimes after the facts.

Detailed comments

• line 43: it is not common to insert references in an abstract; the abstract is also rather long

• line 69: additional references to musical emotion could be added here (see e.g. Juslin & Laukka, 2004; Reybrouck & Eerola, 2017)

• line 132: please explain clearly which are the three aspects of the aesthetic trinity. This is not clear.

• line 138: is it possible to describe the concept of “kama muta” still more in detail, given its importance for this paper and given that the concept is not commonly known

• line 144: the four relationship types are summed up but are not explained. Is it possible to give a very short intuitive description of each of them?

• line 147: what is commensalism (an association between two organisms in which one benefits and the other derives neither benefit nor harm)? what is consubstantial assimilation? Introducing such terms without defining them seems somewhat pedantic and hampers the readability of the paper.

• line 165: do the correlation coefficients refer to either experienced kama muta and trait empathic concern? In that case insert them after these words to avoid confusion. What do the number between square brackets stand for? Is it the interval between the lowest and highest value? the confidence interval? Please indicate this to help the reader. Even for readers with considerable scientific training, the statistics and their abbreviations are sometimes elusive.

• line 183: feeling moved. Why using italics here? Is there something to be stressed?

• line 183: style of referencing: more than three authors: et al.?

• line 192: why semicolon after musical pieces?

• lines 198-199: this sentence is not clear. Is there a hyphen that is lacking?

• line 271: the concept of computing time series seems to be quite challenging. However, after having read the whole paper, it is not yet clear how this is actually done. This should be explained very clearly at the first appearance of the term.

• line 283: same remark. How is the continuous rating done in practice? Not clear. What must the listeners do? Have they to fill is something every second on a list? Must they check boxes or Lickert scale or write down words? This must be explained very clearly from the beginning in order not to affect the readability.

• line 289: it seems that this paragraph relates to the major research questions of the paper. Perhaps this should be stated more explicitly to make the methodology more transparent.

• line 294: as readers we are curious about how this continuous assessment did happen. Please explain.

line 296: what is the motivation behind the selection for these items?

• line 301: here a reference could be made to the mechanisms of thermoregulatory behavior as studied e.g. by Panksepp and others. This is only a suggestion.

• line 305: is there a motivation behind the choice for “emotions perceived” rather than “felt emotions”?

• line 313: please provide a short description of what is meant with the self-scale

• line 315: does it make sense to refer here to a conception of music in terms of agency?

• line 329: this paragraph addresses the previous comment regarding line 289. Perhaps the layout could be changed into a bulleted/numbered list to make the claims more visible as research questions or predictions

• line 332: the second prediction is not clear. What is meant precisely?

• line 343: the difference between empathic concern/fantasy and being moved is not totally clear. Please explain a little more in detail. Does it make sense to refer also to “theory of mind” in this context?

• line 365: explain what is meant here with “cell”? Which cell? What is grouped together in these cells? A short intuitive description is very important to guide the listener. Where do the 7 ratings come from? This is not yet explained and should be done also. Referring to the supplementary material is not enough here? The main text should be understandable also on its own.

• line 370: approval rating: approval of what? Please be clear.

• line 376: it seems as if the reader should have knowledge of the rating here, but up to here, there has not been any description of these ratings, which makes reading and understanding very difficult.

• line 381: same remark as above. Provide at least a short description of the music excerpts. Substitute also “music excerpts” for “excerpts”.

• line 389: this is the answer to the previous comment; perhaps it makes sense to add “see below” at the first referenced to the music excerpt.

• line 398: here at first there is a description of the scales, but this is also not yet sufficiently clear. What are the units of each scale? How are they measured in a continuous way or using a Lickert scale? An example here could be very helpful.

• line 407: here is the answer to the previous remark. This comes too late, or at least there should have been made a reference to this at an earlier place in the text.

line 420: major question: how is this continuous rating done? Please explain the procedure. Is there a time grid? This is not clear?

• line 429: even though I have some considerable knowledge of statistics, it is not clear what the symbols between the brackets stand for: does ‹ mean lesser than or something like this? Please explain a little in intuitive terms what the Interpersonal Reactivity Index is and how we must interpret number given between brackets.

• line: it is not clear how the time series was created. Please explain the timestamped rating procedure. Does this mean that the participant that the subject had to rate on a scale every second? This seems rather hard to do in real time.

• line 447: what is meant with “we decreased the resolution of the time scale”. It seems that this is all very important stuff, but not enough efforts are done to explain the intuitions behind for the non-informed reader. Reference to the supplementary material can help of course, but the paper must be understandable also on its own.

• line 449: increases/decreases over time: what kind of increases/decreases? Please specify.

• line 455: detrended correspondence analysis is a rather technical field, not commonly known to common readers of the journal. Please provide at least some short intuitive description of the technique and how to interpret the results. Explain in intuitive terms what detrending means [the word deterendin can even not be found in an English dictionary], what it is used for and how it offers an additional value for the interpretation of the data. A graphic example could be helpful here, also for explaining the construction of the time series. It seems that such an important aspect of the paper does not come into its own sufficiency by presenting it in a manner that is much too technical and not really understandable. Musicians, and even music psychologist, are not statisticians, mathematicians, or engineers, even if the latter may contribute largely to the computational parts of many papers.

• line 458: same remark. What is “cubic spline detrending”? I am afraid that readers will stop reading here, because of the technicality of the terms. See my suggestions above to insert a box with technical terms and their description.

• line 461: please specify what was the residual method

line 462: what is meant with “pre-registration” here? This seems a strange wording in this context, which is rather confusing.

• line 470: please specify which 11 acoustic features were chosen.

• line 472: two categories: which ones? Please specify them shortly before elaborating on them.

• line 475: here the previous remark regarding line 470 is solved. Perhaps it can help the listener by changing line 472 into “The features are listed below and can typically…”

• line 477: same remark as above; the acoustic features are also rather technical. Yet they are quite important. Perhaps also here a box can be introduced to provide a short description of these terms rather than referring to the supplementary material. This is only a suggestion. In the other case, the terms can be explained more in depth in the supplementary material.

• line 484: here again, the previous remark has been addressed appropriately. Perhaps it makes sense to move this reference to the MITtoolbox already to line 471.

• line 488: Please explain the abbreviation CCF. What does the F stand for?

• line 496: given that the term “lag” is also used in the figures, it makes sense to explain shortly what is meant by this term here. What exactly is “lag zero”? Please explain. This can be done rather easily in intuitive terms as the concept is not so difficult to explain.

• line 506: why the F0 in the CCFO?

• line 559: what does the B score refer to? Please explain. Same remark about sophistication of statistical analysis which may go beyond the competence level of the common readership.

• lines 573 ff: this is very technical section and very difficult to understand for a common reader. It is also very difficult to relate the text and the figure. Please explain much better how to interpret figure 4.

• line 589: same remark about statistical technicality: what does the rs stand for, why three values? Explain a little.

• line 604: same remark: Ward’s minimum variance method. Explain a little.

• line 630: it seems that, after having read the whole paper, very little bas been told about physiological sensations, besides feeling warm in the chest, and it can be asked whether this is a physiological sensation or simple a general bodily feeling. Perhaps “physiological and bodily reactions” may be a better term.

• line 665: same remark

Figures and figure captions:

• Figure 1. Please explain used abbreviations: RE = random-effects? Q = ?: I = ?. The concept of zero lag should also be explained a little so as to make sense for the non-informed reader. A short intuitive description can suffice, either in the caption or in the main text.

• Figure 2. What are the units for the ratings? Please explain in the figure caption. This is done in the supplementary material but not here. Explain the seven point scale shortly.

• Figure 3. Same remark.

• Figure 4. Explain abbreviations. EC = Empathic Concern? It is difficult to relate the text of the figure caption with the figure. Very difficult to understand. Where are the p-values (*, **, ***) in the figure?

• Figure 5: Same remark about intuitive understanding of the figure. How to find and visualize the four clusters. Please provide at least some intuitive explanation. Very technical matter. Difficult to understand.

Suggested additional references

Juslin, P. & P.Laukka (2004). Expression, Perception, and Induction of Musical Emotions: A Review and a Questionnaire Study of Everday Listening. Journal of New Music Research, 33 (3), 217-238.

Reybrouck, M., Eerola, T. (2017). Music and its inductive power: a psychobiological and evolutionary approach to musical emotions. Frontiers in Psychology, 8, Art.No. 494. Open Access

Reviewer #3: This is an interesting study on the correlates of musically elicited states of being moved. Following previous work on this emotional state, the authors preselected their stimuli by either being joyfully or sadly moving, which was confirmed by the continuous ratings provided by the sample. The findings are mostly congruent to the expectations of the authors formulated prior to the study.

The manuscript is well organized and well written. With all this said, I would like to raise a number of theoretical and methodological concerns I see with this work, as well as some minor points.

1. Certain aspects, hypotheses and ideas seem to be not correctly attributed/referenced to the previous literature. For instance, in describing the rating paradigm of the present study, i.e. the continuous self-report, the authors state (line 267) “we aim to investigate experiences of feeling moved in response to music using a paradigm recently employed for studying kama muta evoked by short film clips (Schubert, Zickfeld, Seibt, & Fiske, 2016). Instead of prompting participants to rate their emotional reactions after watching a video, Schubert and colleagues instructed 909 participants to watch a number of different videos online (from a total number of 6) and rate their emotional reactions continuously while watching them.”. The continuous self-report has a long tradition in experimental psychology, reaching back at least to the 1940s (Peterman 1940), a great number of studies and several reviews has been published on that since then (for a recent overview, please see Wagner, Scharinger, Knoop, Menninghaus 2020). The impression of a novel design employed by this study (or that of 2016) is misleading. Similarly, the connection between chills/piloerection and being moved has been first proposed by Konecni (2005) and picked up by Benedek & Kaernbach (2011). First direct empirical evidence was presented by Wassiliwizky et al. (2015).

2. Menninghaus and colleagues’ position on being moved is mentioned repeatedly in the manuscript. However, one of their main points conceptualizing being moved as a mixed emotional state, which is overall always positive, but also includes negative emotions such as sadness, goes unmentioned. Physiological evidence for this position (contraction of corrugator in highly pleasurable moments), was first provided by Wassiliwizky, Koelsch, Wagner, Jacobsen, Menninghaus (2017) and later confirmed by Kimura, Haramizu, Sanada, Oshida (2019). The absence of this aspect in the manuscript is critical, because several result patterns are fully in line with this proposal: for instance, the cross-correlation between being moved and perceived happiness for *both* the sadly and the joyfully moving musical excerpts: CCF = 0.49 [.08, .90] and CCF = 0.61 [.48, .75], respectively. (Overall CCF = 0.67 [.59, .74]). Also note that the happiness and sadness time series in Figure 3 fluctuate in line with the mixed affect proposal. In Wassiliwizky et al. (2015), for instance, the following formula is proposed: For being sadly moved, sadness is in the foreground, but happiness still in the background, whereas for being joyfully moved the reversed picture is true with happiness in the foreground and sadness in the background. Exactly this is reflected by the time courses in Figure 4. Sadness is not 0 (or 1 in this case) for the joyfully moving pieces and reversely happiness is not 0 (or 1 in this case) for the sadly moving pieces.

See also the conclusion in line 684: “When the overall emotional tone of the music was sad rather than joyful, feeling moved or touched cross-correlated with both perceived sadness and perceived joy” These effects appear even for a small number of stimuli and despite the potential bias in the stimulus choice (see point 8 below). Given this situation, I think the interpretation of being moved as a mixed affective state should be offered and discussed in the manuscript.

3. In referring to the literature that links musical chills to the psychoacoustic/musical parameters (p.10/11) the authors might want to add the recent study by Bannister (2020) on the looming effect.

4. The third a priori hypothesis connects being moved to sadness for the sadly moving pieces (line 334: “the time course of feeling moved correlates positively with the time course of perceiving the music as sad for sadly moving excerpts, while it does less so for happily moving excerpts”). How does this fit to the results from 2016 discussed earlier? i.e., “the time series results showed no or even negative associations between these two variables [feeling moved and feeling sad] for most videos (line 280)”.

5. Another question regarding the same third hypothesis, linking being moved to sadness, is: How does this hypothesis relate to the conceptualization of kama muta as a purely positive emotional state (line 148: “kama muta [...] is experienced as primarily positive”)? Why is kama muta, being a positive emotional experience and probed by the label “being moved”, connected to experiences of sadness?

6. It is important to state explicitly that we have a between-subject-design here; it took me quite a while to gain certainty on that due to the somewhat misleading descriptions, e.g. “Participants were presented with all seven excerpts and all seven scales”, line 398. This has far-reaching implications for the interpretation of the time series results. Specifically, we can never conclude that the same person experienced, for instance, being moved and chills at a time. Thus, all interpretations are to be seen on the group level. I see this as a limitation for the present study.

7. Another limitation for me is the fact that participants were asked to provide ratings on perceived --and not felt-- emotions (“we targeted the emotions perceived in the music and not felt emotions”, line 305). Given the strong focus on the experiential aspects in the rest of the paper, I have difficulties to understand this decision. Could the authors provide reasons for their choice of task?

8. Another limitation to be discussed is the experimenter-selected choice of stimuli for this study. It is difficult to rule out the possibility that the experimenters were biased/guided by their concept of kama muta when selecting the stimuli. In the light of the small number of stimuli (see Limitations section) this might be a serious problem, particularly for the analyses on the psychoacoustic correlates, but also for connecting movingness in music to social cohesive effects. I strongly doubt the universality of this link. Symphonic work that has classically been labeled as intensely moving such as the first movements of Beethoven’s fifth symphony or Mahler’s second symphony are only anecdotal references to be mentioned here.

9. Relatedly, I am wondering why “two of the excerpts included lyrics” (line 435), since this study stresses the focus on the link between being moved and instrumental music. Although the languages might be foreign to the perceivers, they can still trigger associations and even more critically convey emotions by the voice. Why could this issue not have been circumvented during stimulus selection? From my perspective, this should at least be discussed as a limitation.

10. It is unusual to have chills ratings as a continuous measure (line 415: “To what degree do you experience chills (goosebumps) right now?”). Usually, they are probed by a binary on-off signal (e.g. pushing a button for the time a chill is experienced). This should be pointed out in the text. Additionally, while I see the advantages of classifying chills as rather mild or strong by this approach, I am also somewhat concerned (at the lower end of the scale) that false positives have been facilitated here. To rule out this concern, I would ask the authors to report the average number/min and duration of chills in their study, i.e. average time stretches between 2/3/4/5 and 1 on their scale. Thus we could compare these values to previous studies that quantify chills/min and the duration of an average chill experience.

11. Relatedly, musically induced chills have always been conceptualized to reflect peak emotional experiences. When we look at the average ratings in Table S3 we see rather low numbers, particularly for the chills themselves. To have a better impression on the local fluctuations I would like to see the continuous average ratings for the physiological variables (chills, warmth in the chest) presented in the same way as the emotional time series in Figures 2 and 3.

12. I sometimes missed the statistically informed justifications for handling the data a certain way in the Data Preparation section. For example, the authors binned the data into 3 seconds units. Why? They refer to their previous paper which used the same strategy. However, this is not a statistical or methodological reason (line 446: “Similar to Schubert and colleagues (2016), we decreased the resolution of the time scales by aggregating judgments within units of three consecutive seconds”).

13. Similarly, out of three different variants of detrending, the authors decided to stick to the spline-detrended data for all further analyses, without providing any (statistical) reasons for this decision. As this deviates from the preregistered procedure, I think this decision should be substantiated.

14. In general, detrending time-series-ratings is reasonable given the effects mentioned in the paper, which would artificially inflate the cross-correlations (i.e., general rating behavior of participants leading to quadratic and cubic trends). However, in such detrending procedures only local/rapid changes survive, whereas long-term trajectories are filtered out and remain unattended. This might be problematic for the rather sluggish signals we are dealing with here (affective fluctuations). Particularly the detrending of chills experiences seems quite counterintuitive to me. I would ask the authors to discuss this issue and also to provide figures with the detrended time series data (in the Supplementary Material), similarly to Figures 2 and 3. At present, the time series we see in Figures 2 and 3 represent raw data that was not the basis for any further analysis.

15. Why only US citizens have been allowed to participate?

16. In justifying the sample size, the authors state (line 364): “Based on recommendations by Schubert, Zickfeld, Seibt, and Fiske (2016) we planned on collecting 40-50 participants per cell”. It is somewhat misleading here to refer to another empirical study and not to statistical methodological work. BTW: In Schubert et al. the sample size of N=40 is already referred to as a previous recommendation, without, however, giving a reference.

17. The assignment of Figures 4 and 5 is mixed up in the text (e.g. line 617: “The averaged feeling moved or touched ratings for each of the clusters can be seen in Figure 5” should be “Figure 4”).

18. The results of the clustering analysis in Figure 4 seem a little bit lost, because the effects could be shown only for two stimuli. I would therefore recommend to interpret them very carefully (currently, we read in line 781 quite generally that empathic concern is associated with more pronounced peaks in continuous feeling moved ratings).

19. The conclusion in line 785 (“these results give support to the role of empathic processes in emotional responses to music (cf. Miu & Vuoskoski, 2017), emphasizing the inherently social nature of musical experience”) seems overgeneralized and overstated given the fact that only moving music has been employed here and the fact that a major study on the functions of music by Schaefer et al. (2013, cited in the manuscript) has identified expression of social relatedness only as a subordinate factor. Ethnographical work on war music and music to intimidate the adversary are further counterexamples to mention here.

20. What are the error bars in Figures 2 and 3?

Reviewer #4: This article describes a thorough and competent piece of research investigating a popular topic in the field of musical emotions (and aesthetics more generally). Nonetheless, there are a few points that I think deserve the authors attention some of which have implications to the interpretation of the results. I hope that these comments help to improve the manuscript.

1. The manuscript in general, and the introduction in particular, could be streamlined. The introduction lacks fluidity because there is a tendency to concatenate the various ideas, concepts, and arguments without carefully crafting the key message. Also, the results description is rather “thick” and could be simplified if the authors focus on the key findings without overinterpreting or trying to explain to much about smaller (sometimes irrelevant) results.

2. Line 165: in the introduction perhaps it is not so relevant to include statistical details, especially because the same is not done for other works as well.

3. The section starting in line 173 would benefit from a description of the work by Coutinho and Scherer (Coutinho, E., & Scherer, K. R. (2017). Introducing the GEneva Music-Induced Affect Checklist (GEMIAC) A Brief Instrument for the Rapid Assessment of Musically Induced Emotions. Music Perception: An Interdisciplinary Journal, 34(4), 371-386.). They did a comparison between the meaning of emotions commonly use to described musical emotion and feeling moved/touched is part of it. What is particularly relevant about their work is that they established the proximity of these feelings to other feelings of emotion, which particularly relevant for this work and perhaps more informative that the sources already cited.

4. Paragraph starting in line 260: First sentence should be further developed to justify the statement. Last sentence also needs a justification (the authors say that they argue the idea but do not present arguments).

5. Line 279: There is evidence that summative and continuous ratings are different things, and the average of continuous ratings does not correspond to summative ones. Therefore, the statement between brackets is not correct. I suggest that the authors develop their ideas in this respect in this part of the manuscript.

6. Lines 305-7: emotions felt and perceived are different. They can be the same, but that also depends on many variables (some of them quite contextual and transient). My point is that the justification presented to used perceived emotions is not clear and needs careful consideration. One may argue that it does not make sense at all compare feeling moved with perceived joy or sadness. In fact, this is a weakness of this work. I suggest that the authors provide stronger rationale for option for perceived emotion ratings and explain in the conclusions that this is acknowledged as a potential limitation in the discussion section.

7. Lines 330-228: text can be simplified. All hypothesis chare the same start. Also, hypotheses III and IV are expressed in a strange way and they seem to overlap. I suggest that you remove the text after the commas in both.

8. Line 419: what are the values between brackets?

9. You collected data that does not seem to have been used in the analysis, especially relating to demographics. Is there a specific reason for that? Could it be that, like trait empathy, these variables could explain differences between subjects? Note as well as there are well established measures of musical proficiency that should have been used.

10. Data preparation: participants continuous ratings are detrended because the of the start and end segments trends. Nonetheless, the authors end up removing the start of the segments which was actually the initial justification for detrending the data. This seems somewhat contradictory. Also, excessive data processing can be very negative as it also eliminates relevant information that you may not be aware of. Intuitively, if the authors are interested on ratings dynamics, any pre-processing that affects that dynamics can have a direct impact in the findings. Can the author explain if using the original data without the start and end segments (which indeed are very much affected by the transition from/to music to/from silence) leads to the same results and conclusions presented in the paper? In my view the paper should report the findings without detrending the data. Unless I am missing something, in which case I would ask the authors to justify more clearly the need for detrending and the consequence that it has on the results and conclusions.

11. Also pertaining to the same point, there seems to be the tendency to use data-processing strategies from another article without further scrutiny. In addition to potentials limitations, it has also led to some inconsistencies. One of them is the fact that the authors ” we decreased the resolution of the time scales by 448 aggregating judgments within units of three consecutive seconds” but used a sliding window (moving average) for acoustic features. This is inconsistent.

12. Can the authors explain why acoustic features were also detrended and what the impact on the data was?

13. Note that figures are of very small quality (at least in the document I receive) and I could not use them as supporting material as I could not read the captions and legends.

14. There is a tendency to overinterpret the results, which sometimes leads to taking conclusions from the analysis of a single piece. A limitation of the study is that it is based on correlations and only a small number of pieces (I perfectly understand that there were reasons to focus only on a few number of pieces) and over interpreting the data can mislead the reader regarding the importance of the findings.

15. I also think that there are interesting trends in the data that are not captured by the correlation analysis. I wonder what the findings would be if the authors would compute the correlations on time series using larger bins (more than 3 seconds) and using overlapping windows to avoid sharp (artificial) changes.

16. Line 564: why did the author use Spearman rank correlation coefficient for acoustic features data and (I suppose) Person’s correlations coefficient for the rest of the data?

17. I could not understand exactly how the authors performed the clustering analysis (line 594 onwards) using the time-series data and why they chose the described the method. Wouldn’t it be possible to use functional ANOVA for this or other well established and robust methods?

18. Results pertaining acoustic features are overinterpreted. The authors neds to keep in mind that there are only 7 tracks and some points being made only pertain to a single track (and the interpretation of the feature meaning is rather subjective). Section starting in line 711 should reflect these changes (as well as in the conclusions). Another important thing that I suggest the authors consider in their interpretation is the fact that correlations between acoustic features and feeling moved may simply be the result of these ratings being correlated with joy or sadness, i.e., if being moved is correlated with perceived joy and perceived joy is correlated with feature X, than we can expect a correlation between being moved and feature X. Nonetheless, we cannot conclude that this correlated in meaningful or that it is not mediated by joy. Correlations must be interpreted carefully.

19. Generally, the discussion and conclusions could be slightly toned down to avoid over interpreting the results of the study.

6. PLOS authors have the option to publish the peer review history of their article (what does this mean?). If published, this will include your full peer review and any attached files.

Reviewer #1: No

Reviewer #2: No

Reviewer #3: No

Reviewer #4: **Yes: **Eduardo Coutinho

---

## [Author Response · Author response to Decision Letter 0]

25 May 2021

Please see our responses in the response letter at the end of the PDF file.

---

## [Decision Letter · Decision Letter 1]

29 Sep 2021

PONE-D-20-21715R1Feeling moved by music: Investigating continuous ratings and acoustic correlatesPLOS ONE

Dear Dr. Vuoskoski,

Thank you for submitting your manuscript to PLOS ONE. After careful consideration, we feel that it has merit but does not fully meet PLOS ONE’s publication criteria as it currently stands. Therefore, we invite you to submit a revised version of the manuscript that addresses the points raised during the review process.

Besides dealing with the reviewer's comment, we would be grateful if you could consider and address, within the paper, the limitations commonly associated with the use of Amazon MTurk/Survey Monkey, namely, the non-naivety and trustworthiness of participants. Please also consider the opportunity of validating the results using another non-web-based sample. For further information, please see http://www.annualreviews.org/doi/abs/10.1146/annurev-clinpsy-021815-093623

and http://journals.plos.org/plosone/article?id=10.1371/journal.pone.0057410#s15.

In alternative, please add a discussion dealing with these potential flaws

We look forward to receiving your revised manuscript.

Kind regards,

Alice Mado Proverbio

Academic Editor

PLOS ONE

Reviewers' comments:

Reviewer's Responses to Questions

**Comments to the Author**

1. If the authors have adequately addressed your comments raised in a previous round of review and you feel that this manuscript is now acceptable for publication, you may indicate that here to bypass the “Comments to the Author” section, enter your conflict of interest statement in the “Confidential to Editor” section, and submit your "Accept" recommendation.

Reviewer #2: All comments have been addressed

Reviewer #3: (No Response)

Reviewer #5: (No Response)

2. Is the manuscript technically sound, and do the data support the conclusions?

Reviewer #2: (No Response)

Reviewer #3: Yes

Reviewer #5: Yes

3. Has the statistical analysis been performed appropriately and rigorously? 

Reviewer #2: (No Response)

Reviewer #3: Yes

Reviewer #5: Yes

4. Have the authors made all data underlying the findings in their manuscript fully available?

Reviewer #2: (No Response)

Reviewer #3: Yes

Reviewer #5: Yes

5. Is the manuscript presented in an intelligible fashion and written in standard English?

Reviewer #2: (No Response)

Reviewer #3: Yes

Reviewer #5: Yes

6. Review Comments to the Author

Reviewer #2: (No Response)

Reviewer #3: The manuscript benefitted from the revision process. Many aspects and details have been corrected, added and/or clarified throughout the text. I am glad that the interpretation is now aligned more closely along the outcomes of the study. I also appreciate that (technical) aspects that could not be addressed or changed have been included into the limitations list.

I have however one remaining concern: the validity of the chills measures. As I already mentioned during the first round, the usual approach is to collect a dichotomous on-off-signal via button presses. My concern was that a continuous measure would inflate the occurrence of chills (which are short, peak-arousal phenomena). Unfortunately, this is exactly the case as we see now with the new additional average durations of 60 s. Most studies report much shorter instances of several chills/min/person (see for instance, Panksepp, 1995; Konecni et al. 2007; Benedek & Kaernbach 2011; Sumpf et al. 2015; Wassiliwizky, Koelsch et al. 2017; Wassiliwizky, Jacobsen et al. 2017; studies by Altenmüller’s group, etc.). I doubt that the authors measured the exactly same phenomenon as the studies they refer to.

Relatedly, I disagree with the statement in the manuscript: “some previous studies have assessed chills using a dichotomous measure probing whether participants experienced chills or not (p.18).

Again, the dichotomous measurement is not an exception but the very standard. In the recent review on chills (which the authors refer to) by de Fleurian & Pearce (2020) it is said: “Most commonly, however, participants report chills by pressing on a button (Bannister, 2020b; Beier et al., 2020; Colver & El-Alayli, 2016; Egermann et al., 2011; Ferreri et al., 2019; Grewe et al., 2011; Grewe et al., 2009a; Grewe et al., 2007; Guhn et al., 2007; Laeng et al., 2016; Mas-Herrero et al., 2014; Mori & Iwanaga, 2014, 2015, 2017; Nagel et al., 2008; Rickard, 2004; Sachs et al., 2016; Salimpoor et al., 2011; Salimpoor et al., 2009; T. W. Schubert et al., 2018; Seibt et al., 2018; Starcke et al., 2019; Sutherland et al., 2009; Wassiliwizky, Koelsch, et al., 2017; Zickfeld, Schubert, Seibt, Blomster, et al., 2019).

I think, this aspect should be addressed thoroughly in the Discussion part of the manuscript.

Reviewer #5: In the present paper, the author present the results of an online study that assessed people’s continuous ratings of musical excerpts with a particular focus on the relationship between ratings of “being moved”.

I understand that this paper has been revised already and I believe that it shows. Clearly, many supplementary analyses have been conducted to back up the main results. My review is based on the revised version alone, without regard to what aspects of the manuscript have been changed or to the responses to previous reviewers.

Overall, I find the paper well-written, the analyses rigorous, and the conclusions justified by the reported results. I particularly appreciate that all results are reported in terms of effect sizes and their confidence intervals. I therefore see no reason to object the publication of this paper given some rather minor changes, detailed below.

1. I think that the logic of pre-determining a musical piece as sadly or joyfully moving needs to be explained earlier on, otherwise, as a naïve reader, I would question the logic of two sections in the introduction:

a. Page 6, “The authors distinguish… “: I cannot follow the logic of tis sentence: Enjoying being moved (which thus needs to depend on being moved because it's a consequence) is only also moderated by being moved if one is sadly moved? How can a cause also moderate its effect?

b. Page 14: III. and IV.: I don't understand how this can be tested without double-dipping into the data

2. P. 20: I find it unbalanced to suggest based on a single paper that time bins of three seconds constitute the temporal building block of perceptual-motor abilities. In fact, it seems very long to me given that much visual perceptual processing occurs at the time scale well below one second (face processing comes to mind especially). Most importantly for the current line of research, however, stable aesthetic ratings have been shown to occur within a second of stimulus presentation (e.g., Belfi et al., 2018; Brielmann, Vale, & Pelli, 2017). I think further justification – or alternatively the demonstration that the particular bin-width of aggregation does not matter as much – is needed.

3. P. 26: Why only the mean and not also the peak? In time series ratings in the aesthetic domain, the peak has been identified as correlating highly with overall evaluations (e.g., Brielmann, Vale, & Pelli, 2017)

4. Figure 6 (especially w regard to its mention on p. 26): to follow along with the interpretation of figure 6 in the text, it would be helpful to indicate which excerpts were joyfully and which ones sadly moving in figure 6 itself

5. P. 28 “strong initial rise within the first 30 seconds”: I think it should be also acknowledged that this might be at least partially dependent on the set starting point in this experiment, i.e., at the lowest end of the scale. So it's not clear whether this is an initial rise in actual responses to the music or just initial hesitation to move the displayed rating.

6. P. 28, second paragraph: based on Figure 1 and the reported results, I cannot see how this statement is correct; cross-correlations appear just as high with perceived beauty and chills, and just as consistent

7. P. 29 “Interestingly, these cross-correlations were consistent…”: There seems to be a missing argument or link between this and the previous sentence: Just because beauty and being moved are positively associated with one another in, overall, sad and joyful pieces does not imply anything about the association between sadness or joyfulness/happiness and beauty

References

Belfi, A. M., Kasdan, A., Rowland, J., Vessel, E. A., Starr, G. G., & Poeppel, D. (2018). Rapid timing of musical aesthetic judgments. Journal of Experimental Psychology: General, 147(10), 1531–1543. doi: 10.1037/xge0000474

Brielmann, A. A., Vale, L., & Pelli, D. G. (2017). Beauty at a glance: The feeling of beauty and the amplitude of pleasure are independent of stimulus duration. Journal of Vision, 17(14), 9. doi: 10.1167/17.14.9

7. PLOS authors have the option to publish the peer review history of their article (what does this mean?). If published, this will include your full peer review and any attached files.

Reviewer #2: No

Reviewer #3: No

Reviewer #5: **Yes: **Aenne Annelie Brielmann

---

## [Author Response · Author response to Decision Letter 1]

16 Nov 2021

Please find our detailed responses to the editor and reviewers in the response letter, which can be found at the end of the PDF file.

---

## [Editor Report · Decision Letter 2]

26 Nov 2021

Feeling moved by music: Investigating continuous ratings and acoustic correlates

PONE-D-20-21715R2

Dear Dr. Vuoskoski,

We’re pleased to inform you that your manuscript has been judged scientifically suitable for publication and will be formally accepted for publication once it meets all outstanding technical requirements. 

Kind regards,

Alice Mado Proverbio

Academic Editor

PLOS ONE

Additional Editor Comments (optional):

Congrats on your nice paper.
---

## [Editor Report · Acceptance letter]

16 Dec 2021

PONE-D-20-21715R2 

Feeling moved by music: Investigating continuous ratings and acoustic correlates 

Dear Dr. Vuoskoski:

I'm pleased to inform you that your manuscript has been deemed suitable for publication in PLOS ONE. Congratulations! Your manuscript is now with our production department. 

Kind regards, 

on behalf of

Dr. Alice Mado Proverbio 

Academic Editor

PLOS ONE